# Multiwavelet-based Operator Learning for Differential Equations

**Gaurav Gupta, Xiongye Xiao, Paul Bogdan**
Ming Hsieh Department of Electrical and Computer Engineering
University of Southern California, Los Angeles, CA 90089
{ggaurav, xiongyex, pbogdan}@usc.edu

## Abstract

The solution of a partial differential equation can be obtained by computing the inverse operator map between the input and the solution space. Towards this end, we introduce a *multiwavelet-based neural operator learning scheme* that compresses the associated operator's kernel using fine-grained wavelets. By explicitly embedding the inverse multiwavelet filters, we learn the projection of the kernel onto fixed multiwavelet polynomial bases. The projected kernel is trained at multiple scales derived from using repeated computation of multiwavelet transform. This allows learning the complex dependencies at various scales and results in a resolution-independent scheme. Compare to the prior works, we exploit the fundamental properties of the operator's kernel which enable numerically efficient representation. We perform experiments on the Korteweg-de Vries (KdV) equation, Burgers' equation, Darcy Flow, and Navier-Stokes equation. Compared with the existing neural operator approaches, our model shows significantly higher accuracy and achieves state-of-the-art in a range of datasets. For the time-varying equations, the proposed method exhibits a $(2X-10X)$ improvement ($0.0018$ ($0.0033$) relative $L2$ error for Burgers' (KdV) equation). By learning the mappings between function spaces, the proposed method has the ability to find the solution of a high-resolution input after learning from lower-resolution data.

## 1 Introduction

Many natural and human-built systems (e.g., aerospace, complex fluids, neuro-glia information processing) exhibit complex dynamics characterized by partial differential equations (PDEs) [52, 60]. For example, the design of wings and airplanes robust to turbulence, requires to learn complex PDEs. Along the same lines, complex fluids (gels, emulsions) are multiphasic materials characterized by a macroscopic behavior [55] modeled by non-linear PDEs. Understanding their variations in viscosity as a function of the shear rate is critical for many engineering projects. Moreover, modeling the dynamics of continuous and discrete cyber and physical processes in complex cyber-physical systems can be achieved through PDEs [68].

Recent efforts on learning PDEs (i.e., mappings between infinite-dimensional spaces of functions), from trajectories of variables, focused on developing machine learning and in particular deep neural networks (NNs) techniques. Towards this end, a stream of work aims at parameterizing the solution map as deep NNs [2, 13, 33, 40, 71]. One issue, however, is that the NNs are tied to a specific resolution during training, and therefore, may not generalize well to other resolutions, thus, requiring retraining (and possible modifications of the model) for every set of discretizations. In parallel, another stream of work focuses on constructing the PDE solution function as a NN architecture [31, 42, 57, 65]. This approach, however, is designed to work with one instance of a PDE and, therefore, upon changing the coefficients associated with the PDE, the model has to be re-trained.

35th Conference on Neural Information Processing Systems (NeurIPS 2021).

Additionally, the approach is not a complete data-dependent one, and hence, cannot be made oblivious to the knowledge of the underlying PDE structure. Finally, the closest stream of work to the problem we investigate is represented by the "Neural Operators" [14, 47, 48, 49, 56]. Being a complete data-driven approach, the neural operators method aims at learning the operator map without having knowledge of the underlying PDEs. The neural operators have also demonstrated the capability of discretization-independence. Obtaining the data for learning the operator map could be prohibitively expensive or time consuming (e.g., aircraft performance to different initial conditions). To be able to better solve the problem of learning the PDE operators from scarce and noisy data, we would ideally explore fundamental properties of the operators that have implications in data-efficient representation.

Our intuition is to transform the problem of learning a PDE to a domain where a compact representation of the operator exists. With a mild assumption regarding the smoothness of the operator's kernel, except finitely many singularities, the multiwavelets [5], with their *vanishing moments property*, sparsify the kernel in their projection with respect to (w.r.t.) a measure. Therefore, learning an operator kernel in the multiwavelet domain is feasible and data efficient. The wavelets have a rich history in signal processing [24, 25], and are popular in audio, image compression [8, 61]. For multiwavelets, the orthogonal polynomial (OP) w.r.t. a measure emerges as a natural basis for the multiwavelet subspace, and an appropriate scale / shift provides a sequence of subspaces which captures the locality at various resolutions. We generalize and exploit the multiwavelets concept to work with arbitrary measures which opens-up new possibilities to design a series of models for the operator learning from complex data streams.

We incorporate the multiwavelet filters derived using a variety of the OP basis into our operator learning model, and show that the proposed architecture outperforms the existing neural operators. Our main contributions are as follows: **(i)** Based on some fundamental properties of the integral operator's kernel, we develop a multiwavelet-based model which learns the operator map efficiently. **(ii)** For the 1-D dataset of non-linear Korteweg-de Vries and Burgers equations, we observe an order of magnitude improvement in the relative $L2$ error (Section 3.1, 3.3). **(iii)** We demonstrate that the proposed model is in validation with the theoretical properties of the pseudo-differential operator (Section 3.2). **(iv)** We show how the proposed multiwavelet-based model is robust towards the fluctuation strength of the input signal (Section 3.1). **(v)** Next, we demonstrate the applicability on higher dimensions of 2-D Darcy flow equation (Section 3.4), and finally show that the proposed approach can learn at lower resolutions and generalize to higher resolutions. The code for reproducing the experiments is available at: `https://github.com/gaurav71531/mwt-operator`.

## 2 Operator Learning using Multiwavelet Transform

We start by defining the problem of operator learning in Section 2.1. Section 2.2 defines the multi-wavelet transform for the proposed operator learning problem and derives the necessary transformation operations across different scales. Section 2.3 outlines the proposed operator learning model. Finally, Section 2.4 lists some of the useful properties of the operators which leads to an efficient implementation of multiwavelet-based models.

### 2.1 Problem Setup

Given two functions $a(x)$ and $u(x)$ with $x \in D$, the operator is a map $T$ such that $Ta = u$. Formally, let $\mathcal{A}$ and $\mathcal{U}$ be two Sobolev spaces $\mathcal{H}^{s,p}$ ($s > 0, p \geq 1$), then the operator $T$ is such that $T : \mathcal{A} \to \mathcal{U}$. The Sobolev spaces are particularly useful in the analysis of partial differential equations (PDEs), and we restrict our attention to $s > 0$ and $p = 2$. Note that, for $s = 0$, the $\mathcal{H}^{0,p}$ coincides with $L^p$, and, $f \in \mathcal{H}^{0,p}$ does not necessarily have derivatives in $L^p$. We choose $p = 2$ in order to be able to define projections with respect to (w.r.t.) measures $\mu$ in a Hilbert space structure.

We take the operator $T$ as an integral operator with the kernel $K : D \times D \to L^2$ such that

$$Ta(x) = \int_D K(x, y)a(y)dy. \tag{1}$$

For the case of inhomogeneous linear PDEs, $\mathcal{L}u = f$, with $f$ being the forcing function, $\mathcal{L}$ is the differential operator, and the associated kernel is commonly termed as Green function. In our case, we do not put the restriction of linearity on the operator. From eq. (1), it is apparent that learning the complete kernel $K(.,.)$ would essentially solve the operator map problem, but it is not necessarily a

numerically feasible solution. Indeed, a better approach would be to exploit possible useful properties (see Section 2.4) such that a compact representation of the kernel can be made. For an efficient representation of the operator kernel, we need an appropriate subspace (or sequence of subspaces), and projection tools to map to such spaces.

**Norm with respect to measures:** Projecting a given function onto a fixed basis would require a measure dependent distance. For two functions $f$ and $g$, we take the inner product w.r.t measure $\mu$ as $\langle f, g \rangle_\mu = \int f(x)g(x)d\mu(x)$, and the associated norm as $||f||_\mu = \langle f, f \rangle_\mu^{1/2}$. We now discuss the next ingredient, which refers to the subspaces required to project the kernel.

## 2.2 Multiwavelet Transform

In this section, we briefly overview the concept of multiwavelets [4] and extend it to work with non-uniform measures at each scale. The multiwavelet transform synergizes the advantages of *orthogonal polynomials* (OPs) as well as the *wavelets* concepts, both of which have a rich history in the signal processing. The properties of wavelet bases like (*i*) vanishing moments, and (*ii*) orthogonality can effectively be used to create a system of coordinates in which a wide class of operators (see Section 2.4) have a *nice* representation. Multiwavelets go few steps further, and provide a fine-grained representation using OPs, but also act as a basis on a finite interval. For the rest of this section, we restrict our attention to the interval $[0, 1]$; however, the transformation to any finite interval $[a, b]$ could be straightforwardly obtained by an appropriate shift and scale.

**Multi Resolution Analysis:** We begin by defining the space of piecewise polynomial functions, for $k \in \mathbb{N}$ and $n \in \mathbb{Z}^+ \cup \{0\}$ as, $\mathbf{V}_n^k = \bigcup_{l=0}^{2^n-1}\{f|\deg(f) < k$ for $x \in (2^{-n}l, 2^{-n}(l+1)) \wedge 0$, elsewhere$\}$. Clearly, $\dim(\mathbf{V}_n^k) = 2^n k$, and for subsequent $n$, each subspace is contained in another as shown by the following relation:

$$\mathbf{V}_0^k \subset \mathbf{V}_1^k \ldots \subset \mathbf{V}_{n-1}^k \subset \mathbf{V}_n^k \subset \ldots. \tag{2}$$

Similarly, we define the sequence of measures $\mu_0, \mu_1, \ldots$ such that $f \in \mathbf{V}_n^k$ is measurable w.r.t. $\mu_n$ and the norm of $f$ is taken as $||f|| = \langle f, f \rangle_{\mu_n}^{1/2}$. Next, since $\mathbf{V}_{n-1}^k \subset \mathbf{V}_n^k$, we define the multiwavelet subspace as $\mathbf{W}_n^k$ for $n \in \mathbb{Z}^+ \cup \{0\}$, such that

$$\mathbf{V}_{n+1}^k = \mathbf{V}_n^k \bigoplus \mathbf{W}_n^k, \quad \mathbf{V}_n^k \perp \mathbf{W}_n^k. \tag{3}$$

For a given OP basis for $\mathbf{V}_0^k$ as $\phi_0, \phi_1, \ldots, \phi_{k-1}$ w.r.t. measure $\mu_0$, a basis of the subsequent spaces $\mathbf{V}_n^k, n > 1$ can be obtained by shift and scale (hence the name, multi-scale) operations of the original basis as follows:

$$\phi_{jl}^n(x) = 2^{n/2}\phi_j(2^n x - l), \quad j = 0, 1, \ldots, k-1, \quad l = 0, 1, \ldots, 2^n - 1, \text{w.r.t.} \quad \mu_n, \tag{4}$$

where, $\mu_n$ is obtained as the collections of shift and scale of $\mu_0$, accordingly.

**Multiwavelets:** For the multiwavelet subspace $\mathbf{W}_0^k$, the orthonormal basis (of piecewise polynomials) are taken as $\psi_0, \psi_1, \ldots, \psi_{k-1}$ such that $\langle \psi_i, \psi_j \rangle_{\mu_0} = 0$ for $i \neq j$ and 1, otherwise. From eq. (3), $\mathbf{V}_n^k \perp \mathbf{W}_n^k$, and since $\mathbf{V}_n^k$ spans the polynomials of degree at most $k$, therefore, we conclude that

$$\int_0^1 x^i \psi_j(x)d\mu_0(x) = 0, \quad \forall 0 \leq j, i < k. \qquad \text{(vanishing moments)} \tag{5}$$

Similarly to eq. (4), a basis for multiwavelet subspace $\mathbf{W}_n^k$ are obtained by shift and scale of $\psi_i$ as $\psi_{jl}^n(x) = 2^{n/2}\psi_j(2^n x - l)$ and $\psi_{jl}^n$ are orthonormal w.r.t. measure $\mu_n$, i.e. $\langle \psi_{jl}^n, \psi_{j'l'}^n \rangle_{\mu_n} = 1$ if $j = j', l = l'$, and 0 otherwise. Therefore, for a given OP basis for $\mathbf{V}_0^k$ (for example, Legendre, Chebyshev polynomials), we only require to compute $\psi_i$, and a complete basis set at all the scales can be obtained using scale/shift of $\phi_i, \psi_i$.

**Note:** Since $\mathbf{V}_1^k = \mathbf{V}_0^k \bigoplus \mathbf{W}_0^k$ from eq. (3), therefore, for a given basis $\phi_i$ of $\mathbf{V}_0^k$ w.r.t. measure $\mu_0$ and $\phi_{jl}^n$ as a basis for $\mathbf{V}_1^k$ w.r.t. $\mu_1$, a set of basis $\psi_i$ can be obtained by applying Gram-Schmidt

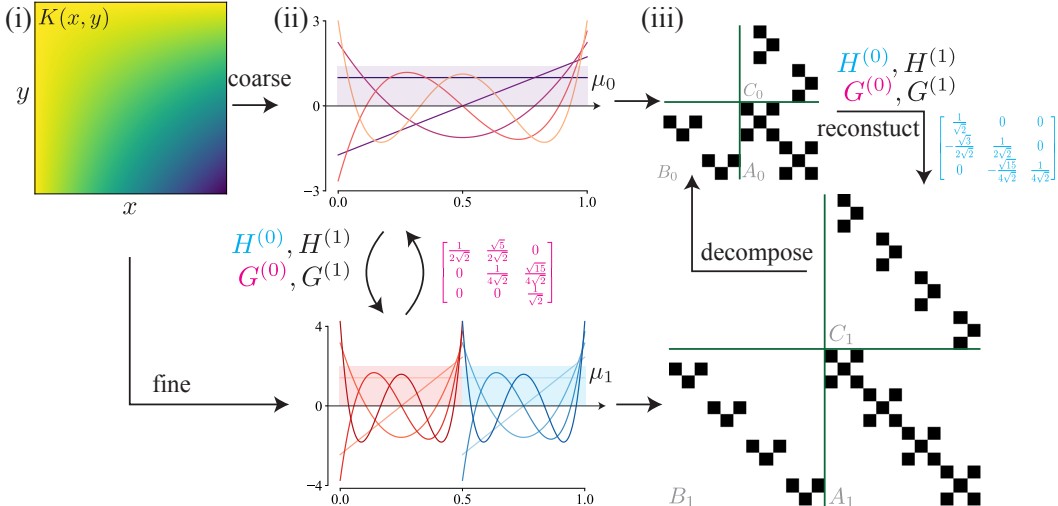

Figure 1: **Multiwavelet representation of the Kernel**. (i) Given kernel $K(x,y)$ of an integral operator $T$, (ii) the bases with different measures $(\mu_0, \mu_1)$ at two different scales (coarse=0, fine=1) projects the kernel into 3 components $A_i, B_i, C_i$. (iii) The decomposition yields a sparse structure, and the entries with absolute magnitude values exceeding $1e^{-8}$ are shown in black. Given projections at any scale, the finer / coarser scale projections can be obtained by reconstruction / decomposition using a fixed multiwavelet filters $H^{(i)}$ and $G^{(i)}, i = 0, 1$.

Orthogonalization using appropriate measures. We refer the reader to supplementary materials for the detailed procedure.

**Note:** Since $\mathbf{V}_0^k$ and $\mathbf{W}_0^k$ lives in $\mathbf{V}_1^k$, therefore, $\phi_i, \psi_i$ can be written as a linear combination of the basis of $V_1^k$. We term these linear coefficients as multiwavelet decomposition filters $(H^{(0)}, H^{(1)}, G^{(0)}, G^{(1)})$, since they are transforming a fine $n = 1$ to coarse scale $n = 0$. A uniform measure ($\mu_0$) version is discussed in [4], and we extend it to any arbitrary measure by including the correction terms $\Sigma^{(0)}$ and $\Sigma^{(1)}$. We refer to supplementary materials for the complete details. The capability of using the non-uniform measures enables us to apply the same approach to any OP basis with finite domain, for example, Chebyshev, Gegenbauer, etc.

For a given $f(x)$, the multiscale, multiwavelet coefficients at the scale $n$ are defined as $\mathbf{s}_l^n = [\langle f, \phi_{il}^n \rangle_{\mu_n}]_{i=0}^{k-1}, \mathbf{d}_l^n = [\langle f, \psi_{il}^n \rangle_{\mu_n}]_{i=0}^{k-1}$, respectively, w.r.t. measure $\mu_n$ with $\mathbf{s}_l^n, \mathbf{d}_l^n \in \mathbb{R}^{k \times 2^n}$. The decomposition / reconstruction across scales is written as

$$\mathbf{s}_l^n = H^{(0)} \mathbf{s}_{2l}^{n+1} + H^{(1)} \mathbf{s}_{2l+1}^{n+1}, \qquad (6) \qquad \mathbf{s}_{2l}^{n+1} = \Sigma^{(0)}(H^{(0)\,T} \mathbf{s}_l^n + G^{(0)\,T} \mathbf{d}_l^n), \qquad (8)$$

$$\mathbf{d}_l^n = G^{(0)} \mathbf{s}_{2l}^{n+1} + H^{(1)} \mathbf{s}_{2l+1}^{n+1}. \qquad (7) \qquad \mathbf{s}_{2l+1}^{n+1} = \Sigma^{(1)}(H^{(1)\,T} \mathbf{s}_l^n + G^{(1)\,T} \mathbf{d}_l^n). \qquad (9)$$

The wavelet (and also multiwavelet) transformation can be straightforwardly extended to multiple dimensions using tensor product of the bases. For our purpose, a function $f \in \mathbb{R}^d$ has multiscale, multiwavelet coefficients $\mathbf{s}_l^n, \mathbf{d}_l^n \in \mathbb{R}^{k \times \dots \times k \times 2^n}$ which are also recursively obtained by replacing the filters in eq. (6)-(7) with their Kronecker product, specifically, $H^{(0)}$ with $H^{(0)} \otimes H^{(0)} \otimes \dots H^{(0)}$, where $\otimes$ is the Kronecker product repeated $d$ times. For eq. (8)-(9) $H^{(0)} \Sigma^{(0)}$ (and similarly others) are replaced with their $d$-times Kronecker product.

**Non-Standard Form:** The multiwavelet representation of the operator kernel $K(x,y)$ can be obtained by an appropriate tensor product of the multiscale and multiwavelet basis. One issue, however, in this approach, is that the basis at various scales are *coupled* because of the tensor product. To untangle the basis at various scales, we use a trick as proposed in [11] called the non-standard wavelet representation. The extra mathematical price paid for the non-standard representation, actually serves as a ground for reducing the proposed model complexity (see Section 2.3), thus, providing data efficiency. For the operator under consideration $T$ with integral kernel $K(x,y)$, let us denote $T_n$ as the projection of $T$ on $V_n^k$, which essentially is obtained by projecting the kernel $K$ onto basis

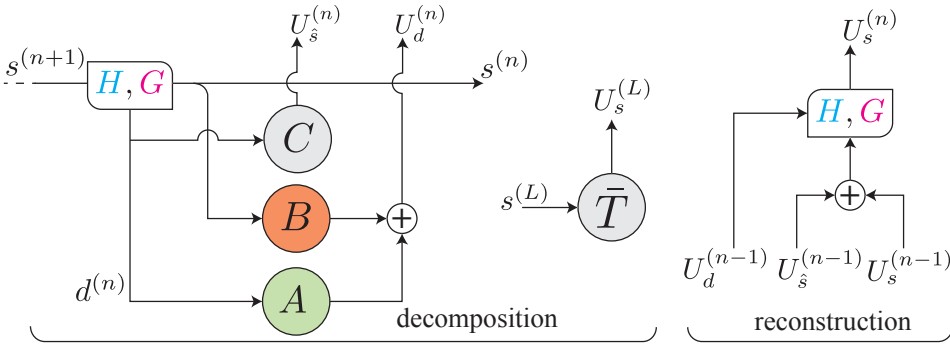

Figure 2: **MWT model architecture**. (**Left**) Decomposition cell using 4 neural networks (NNs) $A, B$ and $C$, and $T$ (for the coarsest scale $L$) performs multiwavelet decomposition from scale $n + 1$ to $n$. (**Right**) Reconstruction module using pre-defined filters $H^{(i)}, G^{(i)}$ performs inverse multiwavelet transform from scale $n - 1$ to $n$.

$\phi_{jl}^n$ w.r.t. measure $\mu_n$. If $P_n$ is the projection operator such that $P_n f = \sum_{j,l} \langle f, \phi_{jl}^n \rangle_{\mu_n} \phi_{jl}^n$, then $T_n = P_n T P_n$. Using telescopic sum, $T_n$ is expanded as

$$T_n = \sum_{i=L+1}^{n} (Q_i T Q_i + Q_i T P_{i-1} + P_{i-1} T Q_i) + P_L T P_L, \qquad (10)$$

where, $Q_i = P_i - P_{i-1}$ and $L$ is the coarsest scale under consideration ($L \geq 0$). From eq. (3), it is apparent that $Q_i$ is the multiwavelet operator. Next, we denote $A_i = Q_i T Q_i, B_i = Q_i T P_{i-1}, C_i = P_{i-1} T Q_i$, and $\bar{T} = P_L T P_L$. In Figure 1, we show the non-standard multiwavelet transform for a given kernel $K(x, y)$. The transformation has a sparse banded structure due to smoothness property of the kernel (see Section 2.4). For the operator $T$ such that $Ta = u$, the map under multiwavelet domain is written as

$$U_{d\,l}^n = A_n d_l^n + B_n s_l^n, \qquad U_{\hat{s}\,l}^n = C_n d_l^n, \qquad U_{s\,l}^L = \bar{T} s_l^L, \qquad (11)$$

where, $(U_{s\,l}^n, U_{d\,l}^n)/(s_l^n, d_l^n)$ are the multiscale, multiwavelet coefficients of $u/a$, respectively, and $L$ is the coarsest scale under consideration. With these mathematical concepts, we now proceed to define our multiwavelet-based operator learning model in the Section 2.3.

## 2.3 Multiwavelet-based Model

Based on the discussion in Section 2.2, we propose a multiwavelet-based model (MWT) as shown in Figure 2. For a given input/output as $a/u$, the goal of the MWT model is to map the multiwavelet-transform of the input $\left(s_l^N\right)$ to output $\left(U_{s\,l}^N\right)$ at the finest scale $N$. The model consists of two parts: (*i*) Decomposition (*dec*), and (*ii*) Reconstruction (*rec*). The *dec* acts as a recurrent network, and at each iteration the input is $s^{n+1}$. Using (6)-(7), the input is used to obtain multiscale and multiwavelet coefficients at a coarser level $s^n$ and $d^n$, respectively. Next, to compute the multiscale/multiwavelet coefficients of the output $u$, we approximate the non-standard kernel decomposition from (11) using four neural networks (NNs) $A, B, C$ and $\bar{T}$ such that $U_{d\,l}^n \approx A_{\theta_A}(d_l^n) + B_{\theta_B}(s_l^n), U_{\hat{s}\,l}^n \approx C_{\theta_C}(d_l^n), \forall\, 0 \leq n < L$, and $U_{s\,l}^L \approx \bar{T}_{\theta_{\bar{T}}}(s_l^L)$. This is a ladder-down approach, and the *dec* part performs the decimation of signal (factor $1/2$), running for a maximum of $L$ cycles, $L < \log_2(M)$ for a given input sequence of size $M$. Finally, the *rec* module collects the constituent terms $U_{s\,l}^n, U_{\hat{s}\,l}^n, U_{d\,l}^n$ (obtained using the *dec* module) and performs a ladder-up operation to compute the multiscale coefficients of the output at a finer scale $n + 1$ using (8)-(9). The iterations continue until the finest scale $N$ is obtained for the output.

At each iteration, the filters in *dec* module downsample the input, but compared to popular techniques (e.g., maxpool), the input is only transformed to a coarser multiscale/multiwavelet space. By virtue of its design, since the non-standard wavelet representation does not have inter-scale interactions, it basically allows us to reuse the same kernel NNs $A, B, C$ at different scales. A follow-up advantage of this approach is that the model is resolution independent, since the recurrent structure of *dec* is input invariant, and for a different input size $M$, only the number of iterations would possibly change for a maximum of $\log_2 M$. The reuse of $A, B, C$ by re-training at various scales also enable us to

learn an expressive model with fewer parameters $(\theta_A, \theta_B, \theta_C, \theta_{\bar{T}})$. We see in Section 3, that even a single-layered CNN for $A, B, C$ is sufficient for learning the operator.

The *dec / rec* module uses the filter matrices which are fixed beforehand, therefore, this part does not require any training. The model does not work for any arbitrary choice of fixed matrices $H, G$. We show in Section 3.4 that for randomly selected matrices, the model does not learn, which validates that careful construction of filter matrices is necessary.

## 2.4 Operators Properties

This section outlines definition of the integral kernels that are typically useful in an efficient compression of the operators through multiwavelets. We then discuss a fundamental property of the pseudo-differential operator.

**Definition 1** ([54]). **Calderón-Zygmund Operator**. *The integral operators that have kernel $K(x, y)$ which is smooth away from the diagonal, and satisfy the following.*

$$
|K(x, y)| \leq \frac{1}{|x - y|},
$$
$$
|\partial_x^M K(x, y)| + |\partial_y^M K(x, y)| \leq \frac{C_0}{|x - y|^{M+1}}.
$$
(12)

The smooth functions with decaying derivatives are *gold* to the multiwavelet transform. Note that, smoothness implies Taylor series expansion, and the multiwavelet transform with sufficiently large $k$ zeroes out the initial $k$ terms of the expansion due to vanishing moments property (5). This is how multiwavelet sparsifies the kernel (see Figure 1 where $K(x, y)$ is smooth). Although, the definition of Calderón-Zygmund is simple (singularities only at the diagonal), but the multiwavelets are capable to compresses the kernel as long as the *number of singularities are finite*.

The next property, from [19], points out that with input/output being single-dimensional functions, for any pseudo-differential operator (with smooth coefficients), the singularity at the diagonal is also well-characterized.

**Property 1.** *Smoothness of Pseudo-Differential Operator. For the integral kernel K(x,y) of a pseudo-differential operator, $K(x, y) \in C^\infty \ \forall x \neq y$, and for $x = y$, $K(x, y) \in C^{T-1}$, where $T + 1$ is the highest derivative order in the given pseudo-differential equation.*

The property 1 implies that, for the class of pseudo-differential operator, and any set of basis with the initial $J$ vanishing moments, the projection of kernel onto such bases will have the diagonal dominating the non-diagonal entries, exponentially, if $J > T - 1$ [19]. For the case of multiwavelet basis with $k$ OPs, $J = k$ (from eq. (5)). Therefore, $k > T - 1$ sparsifies the kernel projection onto multiwavelets, for a fixed number of bits precision $\epsilon$. We see the implication of the Property 1 on our proposed model in the Section 3.2.

## 3 Empirical Evaluation

In this section, we evaluate the multiwavelet-based model (MWT) on several PDE datasets. We show that the proposed MWT model not only exhibits orders of magnitude higher accuracy when compared against the state-of-the-art (Sota) approaches but also works consistently well under different input conditions without parameter tuning. From a numerical perspective, we take the data as point-wise evaluations of the input and output functions. Specifically, we have the dataset $(a_i, u_i)$ with $a_i = a(x_i), u_i = u(x_i)$ for $x_1, x_2, \ldots, x_N \in D$, where $x_i$ are $M$-point discretization of the domain $D$. Unless stated otherwise, the training set is of size 1000 while test is of size 200.

**Model architectures:** Unless otherwise stated, the NNs $A, B$ and $C$ in the proposed model (Figure 2) are chosen as single-layered CNNs following a linear layer, while $\bar{T}$ is taken as single $k \times k$ linear layer. We choose $k = 4$ in all our experiments, and the OP basis as Legendre (Leg), Chebyshev (Chb) with uniform, non-uniform measure $\mu_0$, respectively. The model in Figure 2 is treated as single layer, and for 1D equations, we cascade 2 multiwavelet layers, while for 2D dataset, we use a total 4 layers with $ReLU$ non-linearity.

| Networks | s = 64 | s = 128 | s = 256 | s = 512 | s = 1024 |
|---|---|---|---|---|---|
| MWT Leg | **0.00338** | **0.00375** | **0.00418** | **0.00393** | **0.00389** |
| MWT Chb | 0.00715 | 0.00712 | 0.00604 | 0.00769 | 0.00675 |
| FNO | 0.0125 | 0.0124 | 0.0125 | 0.0122 | 0.0126 |
| MGNO | 0.1296 | 0.1515 | 0.1355 | 0.1345 | 0.1363 |
| LNO | 0.0429 | 0.0557 | 0.0414 | 0.0425 | 0.0447 |
| GNO | 0.0789 | 0.0760 | 0.0695 | 0.0699 | 0.0721 |

Table 1: Korteweg-de Vries (KdV) equation benchmarks for different input resolution $s$. Top: Our methods. Bottom: previous works of Neural operator.

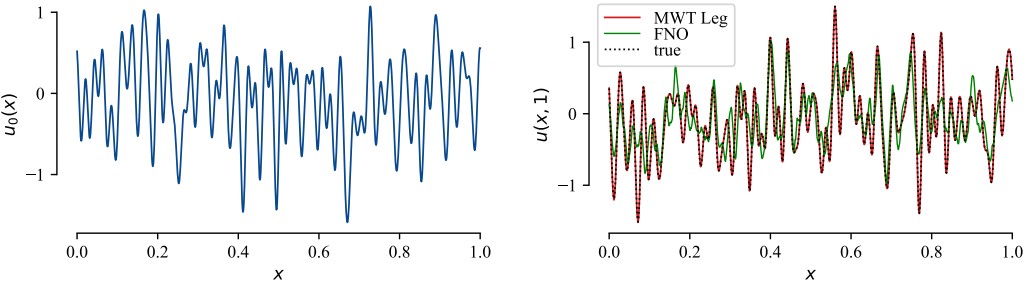

Figure 3: **The output of the KdV equation**. (Left) An input $u_0(x)$ with $\lambda = 0.02$. (Right) The predicted output of the MWT Leg model learning the high fluctuations.

From a mathematical viewpoint, the *dec* and *rec* modules in Figure 2 transform only the multiscale and multiwavelet coefficients. However, the input and output to the model are point-wise function samples, i.e., $(a_i, u_i)$. A remedy around this is to take the data sequence, and construct hypothetical functions $f_a = \sum_{i=1}^{N} a_i \phi_{ji}^n$ and $f_u = \sum_{i=1}^{N} u_i \phi_{ji}^n$. Clearly, $f_a, f_u$ lives in $V_n^k$ with $n = \log_2 N$. Now the model can be used with $s^{(n)} = a_i$ and $U_s^{(n)} = u_i$. Note that $f_a, f_u$ are not explicitly used, but only a matter of convention.

**Benchmark models:** We compare our **MWT** model using two different OP basis (Leg, Chb) with the most recent successful neural operators. Specifically, we consider the graph neural operator (**GNO**) [48], the multipole graph neural operator (**MGNO**) [49], the **LNO** which makes a low-rank ($r$) representation of the operator kernel $K(x, y)$ (also similar to unstacked DeepONet [50]), and the Fourier neural operator (**FNO**) [47]. We experiment on three competent datasets setup by the work of FNO (Burgers' equation (1-D), Darcy Flow (2-D), and Navier-Stokes equation (time-varying 2-D)). In addition, we also experiment with Korteweg-de Vries equation (1-D). For the 1-D cases, a modified FNO with careful parameter selection and removal of Batch-normalization layers results in a better performance compared with the original FNO, and we use it in our experiments. The MWT model demonstrates the highest accuracy in all the experiments. The MWT model also shows the ability to learn the function mapping through lower-resolution data, and able to generalize to higher resolutions.

All the models (including ours) are trained for a total of 500 epochs using Adam optimizer with an initial learning rate (LR) of 0.001. The LR decays after every 100 epochs with a factor of $\gamma = 0.5$. The loss function is taken as relative $L2$ error [47]. All of the experiments are performed on a single Nvidia V100 32 GB GPU, and the results are averaged over a total of 3 seeds.

### 3.1 Korteweg-de Vries (KdV) Equation

The Korteweg-de Vries (KdV) equation was first proposed by Boussinesq [16] and rediscovered by Korteweg and de Vries [23]. KdV is a 1-D non-linear PDE commonly used to describe the non-linear shallow water waves. For a given field $u(x, t)$, the dynamics takes the following form:

$$\frac{\partial u}{\partial t} = -0.5u\frac{\partial u}{\partial x} - \frac{\partial^3 u}{\partial x^3}, x \in (0, 1), t \in (0, 1]$$
$$u_0(x) = u(x, t = 0)$$
(13)

The task for the neural operator is to learn the mapping of the initial condition $u_0(x)$ to the solutions $u(x, t = 1)$. We generate the initial condition in Gaussian random fields according to $u_0 \sim \mathcal{N}(0, 7^4(-\Delta + 7^2 I)^{-2.5})$ with periodic boundary conditions. The equation is numerically solved using *chebfun* package [27] with a resolution $2^{10}$, and datasets with lower resolutions are obtained by sub-sampling the highest resolution data set.

**Varying resolution:** The experimental results of the KdV equation for different input resolutions $s$ are shown in Table 1. We see that, compared to any of the benchmarks, our proposed MWT Leg exhibits the lowest relative error and is lowest nearly by an order of magnitude. Even in the case of the resolution of 64, the relative error is low, which means that a sparse data set with a coarse resolution of 64 is sufficient for the neural operator to learn the function mapping between infinite-dimensional spaces.

**Varying fluctuations:** We now vary the smoothness of the input function $u_0(x, 0)$ by controlling the parameter $\lambda$, where low values of $\lambda$ imply more frequent fluctuations and $\lambda \to 0$ reaches the Brownian motion limit [30]. To isolate the importance of incorporating the multiwavelet transformation, we use the same convolution operation as in FNO, i.e., Fourier transform-based convolution with different modes $k_m$ (only single-layer) for $A, B, C$. We see in Figure 4 that MWT model consistently outperforms the recent baselines for all the values of $\lambda$. A sample input/output from test set is shown in the Figure 3. The FNO model with higher values of $k_m$ has better performance due to more Fourier bases for representing the high-frequency signal, while MWT does better even with low modes in its $A, B, C$ CNNs, highlighting the importance of using wavelet-based filters in the signal processing.

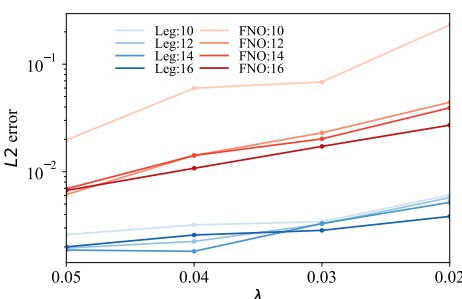

Figure 4: Comparing MWT by varying the degree of fluctuations $\lambda$ in the input with resolution $s = 1024$. For each convolution, we fix the number of Fourier bases as $k_m$. For FNO, the width is 64.

## 3.2 Theoretical Properties Validation

We test the ability of the proposed MWT model to capture the theoretical properties of the pseudo-differential operator in this Section. Towards that, we consider the Euler-Bernoulli equation [62] that models the vertical displacement of a finite length beam over time. A Fourier transform version of the beam equation with the constraint of both ends being clamped is as follows

$$\frac{\partial^4 u}{\partial x^4} - \omega^2 u = f(x), \quad \frac{\partial u}{\partial x}\bigg|_{x=0}^{x=1} = 0 \tag{14}$$
$$u(0) = u(1) = 0,$$

where $u(x)$ is the Fourier transform of the time-varying beam displacement, $\omega$ is the frequency, $f(x)$ is the applied force. The Euler-Bernoulli is a pseudo-differential equation with the maximum derivative order $T + 1 = 4$. We take the task of learning the map from $f$ to $u$. In Figure 5, we see that for $k \geq 3$, the models relative error across epochs is similar, however, they are different for $k < 3$, which is in accordance with the Property 1. For $k < 3$, the multiwavelets will not be able to annihilate the diagonal of the kernel which is $C^{T-1}$, hence, sparsification cannot occur, and the model learns slow.

## 3.3 Burgers' Equation

The 1-D Burgers' equation is a non-linear PDE occurring in various areas of applied mathematics. For a given field $u(x, t)$ and diffusion coefficient $v$, the 1-D Burgers' equation reads:

$$\frac{\partial u}{\partial t} = -u\frac{\partial u}{\partial x} + v\frac{\partial^2 u}{\partial x^2}, x \in (0, 2\pi), t \in (0, 1] \tag{15}$$
$$u_0(x) = u(x, t = 0).$$

The task for the neural operator is to learn the mapping of initial condition $u(x, t = 0)$ to the solutions at $t = 1$ $u(x, t = 1)$. To compare with many advanced neural operators under the same conditions,

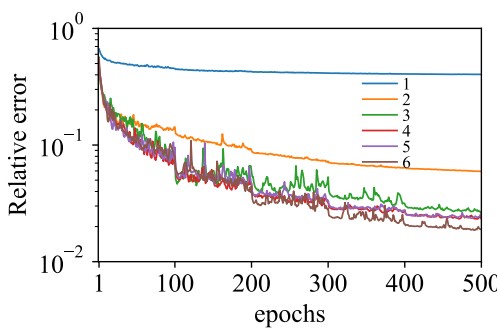

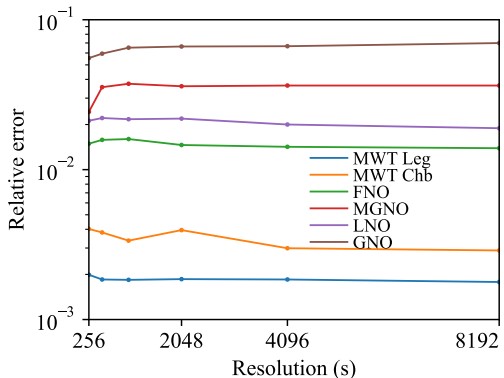

Figure 5: Relative $L2$ error vs epochs for MWT Leg with different number of OP basis $k = 1, \ldots, 6$.

Figure 6: Burgers' Equation validation at various input resolution $s$. Our methods: MWT Leg, Chb.

| Networks | s = 32 | s = 64 | s = 128 | s = 256 | s = 512 |
|----------|--------|--------|---------|---------|---------|
| MWT Leg | **0.0152** | **0.00899** | **0.00747** | **0.00722** | **0.00654** |
| MWT Chb | 0.0174 | 0.0108 | 0.00872 | 0.00892 | 0.00891 |
| MWT Rnd | 0.2435 | 0.2434 | 0.2434 | 0.2431 | 0.2432 |
| FNO | 0.0177 | 0.0121 | 0.0111 | 0.0107 | 0.0106 |
| MGNO | 0.0501 | 0.0519 | 0.0547 | 0.0542 | - |
| LNO | 0.0524 | 0.0457 | 0.0453 | 0.0428 | - |

Table 2: Benchmarks on Darcy Flow equation at various input resolution $s$. Top: Our methods. MWT Rnd instantiate random entries of the filter matrices in (6)-(9). Bottom: prior works on Neural operator.

we use the Burgers' data and the results that have been published in [47] and [49]. The initial condition is sampled as Gaussian random fields where $u_0 \sim \mathcal{N}(0, 5^4(-\Delta + 5^2 I)^{-2})$ with periodic boundary conditions. $\Delta$ is the Laplacian, meaning the initial conditions are sampled by sampling its first several coefficients from a Gaussian distribution. In the Burgers' equation, $v$ is set to $0.1$. The equation is solved with resolution $2^{13}$, and the data with lower resolutions are obtained by sub-sampling the highest resolution data set.

The results of the experiments on Burgers' equation for different resolutions are shown in Figure 6. Compared to any of the benchmarks, our MWT Leg obtains the lowest relative error, which is an order of magnitude lower than the state-of-the-art. It's worth noting that even in the case of low resolution, MWT Leg still maintains a very low error rate, which shows its potential for learning the function mapping through low-resolution data, that is, the ability to map between infinite-dimensional spaces by learning a limited finite-dimensional spaces mapping.

### 3.4 Darcy Flow

Darcy flow formulated by Darcy[22] is one of the basic relationships of hydrogeology, describing the flow of a fluid through a porous medium. We experiment on the steady-state of the 2-d Darcy flow equation on the unit box, where it takes the following form:

$$\nabla \cdot (a(x)\nabla u(x)) = f(x), \quad x \in (0,1)^2$$
$$u(x) = 0, \quad x \in \partial(0,1)^2 \tag{16}$$

We set the experiments to learn the operator mapping the coefficient $a(x)$ to the solution $u(x)$. The coefficients are generated according to $a \sim \mathcal{N}(0, (-\Delta + 3^2 I)^{-2})$, where $\Delta$ is the Laplacian with zero Neumann boundary conditions. The threshold of $a(x)$ is set to achieve ellipticity. The solutions $u(x)$ are obtained by using a 2nd-order finite difference scheme on a $512 \times 512$ grid. Data sets of lower resolution are sub-sampled from the original data set.

The results of the experiments on Darcy Flow for different resolutions are shown in Table2. MWT Leg again obtains the lowest relative error compared to other neural operators at various resolutions.

We also perform an additional experiment, in which the multiwavelet filters $H^{(i)}, G^{(i)}, i = 0, 1$ are replaced with random values (properly normalized). We see in Table 2, that MWT Rnd does not learn the operator map, in fact, its performance is worse than all the models. This signifies the importance of the careful choice of the filter matrices.

### 3.5 Additional Experiments

Full results for these experiments are provided in the supplementary materials.

**Navier Stokes Equation:** The Navier-Stokes (NS) are 2d time-varying PDEs modeling the viscous, incompressible fluids. The proposed MWT model does a 2d multiwavelet transform for the velocity $u$, while uses a single-layered 3d convolution for $A, B$ and $C$ to learn dependencies across space-time. We have observed that the proposed MWT Leg is in par with the Sota on the NS equations.

**Prediction at high resolution:** We show that MWT model trained at lower resolutions for various datasets (for example, training with $s = 256$ for Burgers) can predict the output at finer resolutions $s = 2048$, with relative error of $0.0226$, thus eliminating the need for expensive sampling. The training and testing with $s = 2048$ yields a relative error of $0.00189$.

**Train/evaluation with different sampling rules:** We study the operator learning behavior when the training and evaluation datasets are obtained using the random function from different generating rules. The training is done with squared exponential kernel but evaluation is done on different generating rule [30] with controllable parameter $\lambda$.

## 4 Conclusion

We address the problem of data-driven learning of the operator that maps between two function spaces. Motivated from the fundamental properties of the integral kernel, we found that multiwavelets constitute a natural basis to represent the kernel sparsely. After generalizing the multiwavelets to work with arbitrary measures, we proposed a series of models to learn the integral operator. This work opens up new research directions and possibilities toward designing efficient Neural operators utilizing properties of the kernels, and the suitable basis. We anticipate that the study of this problem will solve many engineering and biological problems such as aircraft wing design, complex fluids dynamics, metamaterials design, cyber-physical systems, neuron-neuron interactions that are modeled by complex PDEs.

## Acknowledgement

We are thankful to the anonymous reviewers for providing their valuable feedback which improved the manuscript. We would also like to thank Radu Balan for his valuable feedback. We gratefully acknowledge the support by the National Science Foundation Career award under Grant No. CPS/CNS-1453860, the NSF award under Grant CCF-1837131, MCB-1936775, CNS-1932620, the U.S. Army Research Office (ARO) under Grant No. W911NF-17-1-0076, the Okawa Foundation award, and the Defense Advanced Research Projects Agency (DARPA) Young Faculty Award and DARPA Director Award under Grant No. N66001-17-1-4044, an Intel faculty award and a Northrop Grumman grant. A part of this work used the Extreme Science and Engineering Discovery Environment (XSEDE), which is supported by National Science Foundation grant number ACI-1548562. The views, opinions, and/or findings contained in this article are those of the authors and should not be interpreted as representing the official views or policies, either expressed or implied by the Defense Advanced Research Projects Agency, the Army Research Office, the Department of Defense or the National Science Foundation.

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
