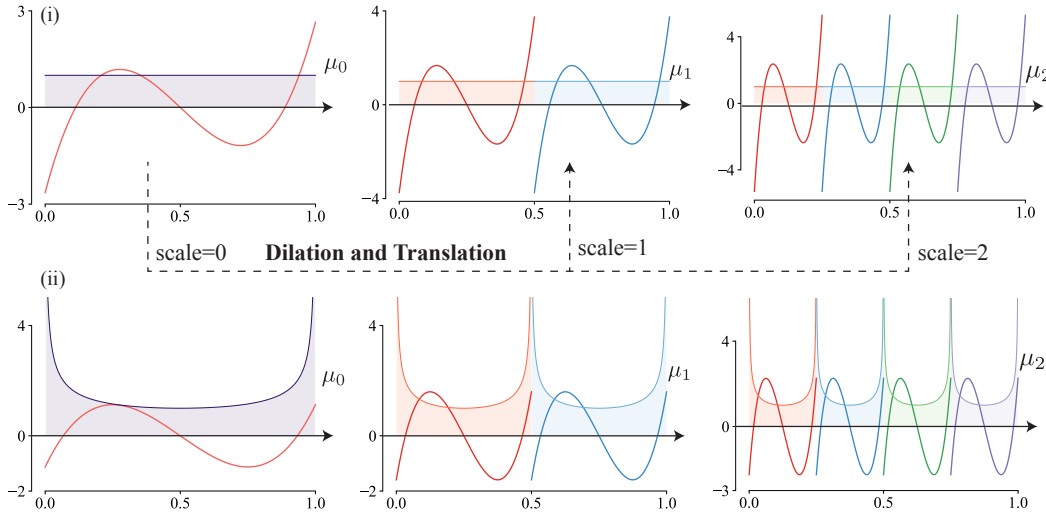

Figure 7: **Wavelet dilation and translation**. The dilation and translation of the mother wavelet function from left to right. The scale$= 0$ represents the mother wavelet function with its measure $\mu_0$. The higher scales $(1, 2)$ are obtained by scale/shift with a factor of 2. **(i)** Mother wavelet using shifted Legendre polynomial $P_3(2x - 1)$ with the uniform measure $\mu_0$, while **(ii)** uses shifted Chebyshev polynomial $T_3(2x - 1)$ with the non-uniform measure $\mu_0$.

## A    Technical Background

We present here some technical preliminaries that are used in the current work. The literature for some of the topics is vast, and we list only the properties that are useful specifically for this work.

### A.1    Wavelets

The wavelets represent sets of functions that result from dilation and translation from a single function, often termed as 'mother function', or 'mother wavelet'. For a given mother wavelet $\psi(x)$, the resulting wavelets are written as

$$\psi_{a,b}(x) = \frac{1}{|a|^{1/2}} \psi \left( \frac{x - b}{a} \right), \qquad a, b \in \mathbb{R}, a \neq 0, x \in D, \tag{17}$$

where $a, b$ are the dilation, translation factor, respectively, and $D$ is the domain of the wavelets under consideration. In this work, we are interested in the compactly supported wavelets, or $D$ is a finite interval $[l, r]$, and we also take $\psi \in L^2$. The consideration for non-compact wavelets, for example, Daubechies [26] will be a future consideration. For the rest of this work, without loss of generality, we restrict ourself to the finite domain $D = [0, 1]$, and extension to any $[l, r]$ can be simply done by making suitable shift and scale.

From a numerical perspective, discrete values (or Discrete Wavelet Transform) of $a, b$ are more useful, and hence, we take $a = 2^{-j}, j = 0, 1, \ldots, L - 1$, where $L$ are the finite number of scales up to which the dilation occurs, and the dilation factor is 2. For a given value of $a = 2^{-j}$, the values of $b$ can be chosen as , $b = na, \quad n = 0, 1, \ldots, 2^j - 1$. The resulting wavelets are now expressed as $\psi_{j,n}(x) = 2^{j/2}\psi(2^j x - n), \quad n = 0, 1, \ldots 2^j - 1$, and $x \in [n2^{-j}, (n + 1)2^{-j}]$. Given a mother wavelet function, the dilation and translation operations for three scales $(L = 3)$ is shown in Figure 7. For a given function $f$, the discrete wavelet transform is obtained by projecting the function $f$ onto the wavelets $\psi_{j,n}$ as

$$c_{j,n} = \int_{n2^{-j}}^{(n+1)2^{-j}} f(x)\psi_{j,n}dx, \tag{18}$$

where $c_{j,n}$ are the discrete wavelet transform coefficients.

## A.2 Orthogonal Polynomials

The next set of ingredients that are useful to us are the family of orthogonal polynomials (OPs). Specifically, the OPs in the current work will serve as the mother wavelets or span the 'mother subspace' (see Section A.3). Therefore, we are interested in the OPs that are non-zero over a finite domain, and are zero almost everywhere (a.e.). For a given measure $\mu$ that defines the OPs, a sequence of OPs $P_0(x), P_1(x), \ldots$ satisfy $\deg(P_i) = i$, and $\langle P_i, P_j \rangle_\mu = 0, \forall i \neq j$, where $\langle P_i, P_j \rangle_\mu = \int P_i(x) P_j(x) d\mu$. Therefore, sequence of OPs are particularly useful as they can act as a set of basis for the space of polynomials with degree $< d$ by using $P_0, \ldots, P_{d-1}(x)$.

The popular set of OPs are hypergeometric polynomials (also known as Jacobi polynomials). Among them, the common choices are Legendre, Chebyshev, and Gegenbauer (which generalize Legendre and Chebyshev) polynomials. These polynomials are defined on a finite interval of $[-1, 1]$ and are useful for the current work. The other set of OPs are Laguerre, and Hermite polynomials which are defined over non-finite domain. Such OPs could be used for extending the current work to non-compact wavelets. We now review some defining properties of the Legendre and Chebyshev polynomials.

### A.2.1 Legendre Polynomials

The Legendre polynomials are defined with respect to (w.r.t.) a uniform weight function $w_L(x) = 1$ for $-1 \leq x \leq 1$ or $w_L(x) = \mathbf{1}_{[-1,1]}(x)$ such that

$$\int_{-1}^{1} P_i(x) P_j(x) dx = \begin{cases} \frac{2}{2i+1} & i = j, \\ 0 & i \neq j. \end{cases} \tag{19}$$

For our purpose, we shift and scale the Legendre polynomials such that they are defined over $[0, 1]$ as $P_i(2x - 1)$, and the corresponding weight function as $w_L(2x - 1)$.

**Derivatives:** The Legendre polynomials satisfy the following recurrence relationships

$$iP_i(x) = (2i - 1)xP_{i-1}(x) - (i - 1)P_{i-2}(x),$$
$$(2i + 1)P_i(x) = P'_{i+1}(x) - P'_{i-1}(x),$$

which allow us to express the derivatives as a linear combination of lower-degree polynomials itself as follows:

$$P'_i(x) = (2i - 1)P_{i-1}(x) + (2i - 3)P_{i-1}(x) + \ldots, \tag{20}$$

where the summation ends at either $P_0(x)$ or $P_1(x)$, with $P_0(x) = 1$ and $P_1(x) = x$.

**Basis:** A set of orthonormal basis of the space of polynomials with degree $< d$ defined over the interval $[0, 1]$ is obtained using shifted Legendre polynomials such that

$$\phi_i = \sqrt{2i + 1} P_i(2x - 1),$$

w.r.t. weight function $w(x) = w_L(2x - 1)$, such that

$$\langle \phi_i, \phi_j \rangle_\mu = \int_0^1 \phi_i(x) \phi_j(x) dx = \delta_{ij}.$$

### A.2.2 Chebyshev Polynomials

The Chebyshev polynomials are two sets of polynomial sequences (first, second order) as $T_i, U_i$. We take the polynomial of the first order $T_i(x)$ of degree $i$ which is defined w.r.t. weight function $w_{Ch}(x) = 1/\sqrt{1 - x^2}$ for $-1 \leq x \leq 1$ as

$$\int_{-1}^{1} T_i(x) T_j(x) \frac{1}{\sqrt{1 - x^2}} dx = \begin{cases} \pi & i = j = 0, \\ \pi/2 & i = j > 0, \\ 0 & i \neq j. \end{cases} \tag{21}$$

After applying the scale and shift to the Chebyshev polynomials such that their domain is limited to $[0, 1]$, we get $T_i(2x - 1)$ and the associated weight function as $w_{Ch}(2x - 1)$ such that $T_i(2x - 1)$ are orthogonal w.r.t. $w_{Ch}(2x - 1)$ over the interval $[0, 1]$.

**Derivatives:** The Chebyshev polynomials of the first order satisfy the following recurrence relationships

$$2T_i(x) = \frac{1}{i+1}T'_{i+1}(x) - \frac{1}{i-1}T'_{i-1}(x), \quad i > 1,$$
$$T_{i+1}(x) = 2xT_i(x) - T_{i-1}(x),$$

The derivative of the $T_i(x)$ can be written as the following summation of sequence of lower degree polynomials

$$T'_i(x) = i(2T_{i-1}(x) + 2T_{i-3}(x) + \ldots),$$

where the series ends at either $T_0(x) = 1$, or $T_1(x) = x$. Alternatively, the derivative of $T_i(x)$ can also be written as $T'_i(x) = iU_{i-1}(x)$, where $U_i(x)$ is the second-order Chebyshev polynomial of degree $i$.

**Basis:** A set of orthonormal basis of the space of polynomials of degree up to $d$ and domain $[0, 1]$ is obtained using Chebyshev polynomials as

$$\phi_i = \begin{cases} \frac{2}{\sqrt{\pi}}T_i(2x-1) & i > 0, \\ \sqrt{\frac{2}{\pi}} & i = 0. \end{cases}$$

w.r.t. weight function $w_{Ch}(2x - 1)$, or

$$\langle \phi_i, \phi_j \rangle_\mu = \int_0^1 \phi_i(x)\phi_j(x)w_{Cb}(2x-1)dx = \delta_{ij}.$$

**Roots:** Another useful property of Chebyshev polynomials is that they can be expressed as trigonometric functions; specifically, $T_n(\cos\theta) = \cos(n\theta)$. The roots of such are also well-defined in the interval $[-1, 1]$. For $T_n(x)$, the $n$ roots $x_1, \ldots, x_n$ are given by

$$x_i = \cos\left(\frac{\pi}{n}\left(i - \frac{1}{2}\right)\right).$$

### A.3 Multiwavelets

The multiwavelets, as introduced in [6], exploit the advantages of both wavelets (Section A.1) as well as OPs (Section A.2). For a given function $f$, instead of projecting the function onto a single wavelet function (wavelet transform), the multiwavelets go one step further and projects the function onto a subspace of degree-restricted polynomials. Along the essence of the wavelet-transform, in multiwavelets, a sequence of wavelet bases are constructed which are scaled/shifted version of the basis of the coarsest scale polynomial subspace.

In this work, we present a measure-version of the multiwavelets which opens-up a family of the multiwavelet-based models for the operator learning. In Section C.1, we provide a detailed mathematical formulation for developing multiwavelets using any set of OPs with measures which can be non-uniform. To be able to develop compactly supported multiwavelets, we have restricted ourself to the family of OPs which are non-zero only over a finite interval. The extension to non-compact wavelets could be done by using OPs which are non-zero over complete/semi range of the real-axis (for example, Laguerre, Hermite polynomials). As an example, we present the expressions for Legendre polynomials which use uniform measure in Section C.2, and Chebyshev polynomials which use non-uniform measure in Section C.3. The work can be readily extended to other family of OPs like Gegenbauer polynomials.

### A.4 Pseudo-Differential Equations

The linear inhomogeneous pseudo-differential equations $\mathcal{L}u = f$ have the operator which takes the following form

$$\mathcal{L} = \sum_{\alpha \in \mathbb{A}} a_\alpha(x)\partial_x^\alpha, \tag{22}$$

where $\mathbb{A}$ is the subset of natural numbers $\mathbb{N} \cup \{0\}$, and $x \in \mathbb{R}^n$. The order of the equation is denoted by the highest integer in the set $\mathbb{A}$. The simplest and the most useful case of pseudo-differential operators $\mathcal{L}$ is the one in which $a_\alpha(x) \in C^\infty$. In the pseudo-differential operators literature, it is often convenient to have a symbolic representation for the pseudo-differential operator. First, the Fourier transform of a function $f$ is taken as $\hat{f}(\xi) = \int_{\mathbb{R}^n} f(x) e^{-i2\pi\xi x} dx$. The pseudo-differential operator over a function $f$ is defined as

$$T_a f(x) = \int_{\mathbb{R}^n} a(x, \xi) e^{i2\pi\xi x} \hat{f}(x) dx,$$

where the operator $T_a$ is parameterized by the symbol $a(x, \xi)$ which for the differential equation (22) is given by

$$a(x, \xi) = \sum_{\alpha \in \mathbb{A}} a_\alpha(x) (2\pi i \xi)^\alpha.$$

The Euler-Bernoulli equation discussed in the Section 3.2 has $\mathbb{A} = \{0, 4\}$.

# B  Multiwavelet Filters

We discuss in details the multiwavelet filters as presented in the Section 2.2. First, we introduce some mathematical terminologies, specifically, useful for multiwavelets filters in Section B.1, and then preview few useful tools in Sections B.2, B.3.

## B.1  Measures, Basis, and Projections

**Measures:** The functions are expressed w.r.t. basis usually by using measures $\mu$ which could be non-uniform in-general. Intuitively, the measure provides weights to different locations over which the specified basis are defined. For a measure $\mu$, let us consider a Radon-Nikodym derivative as $w(x) := \frac{d\mu}{d\lambda}(x)$, where, $d\lambda := dx$ is the Lebesgue measure. In other words, the measure-dependent integrals $\int f d\mu(x)$, can now be defined as $\int f(x) w(x) dx$.

**Basis:** A set of orthonormal basis w.r.t. measure $\mu$, are $\phi_0, \ldots, \phi_{k-1}$ such that $\langle \phi_i, \phi_j \rangle_\mu = \delta_{ij}$. With the weighting function $w(x)$, which is a Radon-Nikodym derivative w.r.t. Lebesgue measure, the orthonormality condition can be re-written as $\int \phi_i(x) \phi_j(x) w(x) dx = \delta_{ij}$.

The basis can also be appended with a multiplicative function called *tilt* $\chi(x)$ such that for a set of basis $\phi_i$ which is orthonormal w.r.t. $\mu$ with weighting function $\frac{d\mu}{d\lambda}(x) = w(x)$, a new set of basis $\phi_i \chi$ are now orthonormal w.r.t. a measure having weighting function $w/\chi^2$. We will see that for OPs like Chebyshev in Section C.3, a proper choice of tilt $\chi(x)$ simplifies the analysis.

**Projections:** For a given set of basis $\phi_i$ defined w.r.t. measure $\mu$ and corresponding weight function $w(x)$, the inner-products are defined such that they induce a measure-dependent Hilbert space structure $\mathcal{H}_\mu$. Next, for a given function $f$ such that $f \in \mathcal{H}_\mu$, the projections onto the basis polynomials are defined as $c_i = \int f(x) \phi_i(x) w(x) dx$.

## B.2  Gaussian Quadrature

The Gaussian quadrature are the set of tools which are useful in approximating the definite integrals of the following form

$$\int_a^b f(x) w(x) dx \approx \sum_{i=1}^n \omega_i f(x_i), \tag{23}$$

where, $\omega_i$ are the scalar weight coefficients, and $x_i$ are the $n$ locations chosen appropriately. For a $n$-point quadrature, the eq. (23) is exact for the functions $f$ that are polynomials of degree $\leq 2n - 1$. This is particularly useful to us, as we see in the Section C.

From the result in [64], it can be argued that, for a class of OPs $P_i$ defined w.r.t. weight function $w(x)$ over the interval $[a, b]$ such that $x_1, x_2, \ldots, x_n$ are the roots of $P_n$, if

$$\sum_{i=1}^n \omega_i P_k(x_i) = \begin{cases} ||P_0||_\mu^2 & k = 0, \\ 0 & k > 0, \end{cases}$$

then,

$$\sum_{i=1}^{n} \omega_i f(x_i) = \int_a^b f(x)w(x)dx,$$

for any $f$ such that $f$ is a polynomial of degree $\leq 2n-1$. The weight coefficients can also be written in a closed-form expression [1] as follows

$$\omega_i = \frac{a_n}{a_{n-1}} \frac{\int_a^b P_{n-1}^2(x)w(x)dx}{P_n'(x_i)P_{n-1}(x_i)}, \tag{24}$$

where, $a_n$ is the coefficient of $x^n$ in $P_n$. Thus, the integral in (23) can be computed using family of OPs defined w.r.t. weight function $w(x)$. Depending on the class of OPs chosen, the Gaussian quadrature formula can be derived accordingly using eq. (24). For a common choice of OPs, the corresponding name for the Quadrature is 'Gaussian-Legendre', 'Gaussian-Chebyshev', 'Gaussian-Laguerre', etc.

### B.3   Gram-Schmidt Orthogonalization

The Gram-Schmidt Orthogonalization (GSO) is a common technique for deriving a (i) set of vectors in a subspace, orthogonal to an (ii) another given set of vectors. We briefly write the GSO procedure for obtaining a set of orthonormal polynomials w.r.t. measures which in-general is different for polynomials in set (i) and (ii). Specifically, we consider that for a given subspace of polynomials with degree $< k$ as $V_0$ and another subspace of polynomials with degree $< k$ $V_1$, such that $V_0 \subset V_1$, we wish to obtain a set of orthonormal basis for the subspace of polynomials with degree $< k$ $W_0$, such that $V_0 \perp W_0$ and $W_0 \subset V_1$. It is apparent that, if $\dim(W_0) = n$, $\dim(V_0) = m$ and $\dim(V_1) = p$, then $m + n \leq p$.

Let $(\psi_0, \ldots, \psi_{n-1})$ be a set of basis of the polynomial subspace $W_0$, $(\phi_0^{(0)}, \ldots, \phi_{m-1}^{(0)})$ be a set of basis for $V_0$, and $(\phi_1^{(0)}, \ldots, \phi_{p-1}^{(1)})$ be a set of basis for $V_1$. We take that basis $\psi_i$ and $\phi_i^{(0)}$ are defined w.r.t. same measure $\mu_0$, while $\phi_i^{(1)}$ are defined w.r.t. a different measure $\mu_1$. A set of $\psi_i$ can be obtained by iteratively applying the following procedure for $i = 0, 1, \ldots, n-1$

$$\begin{aligned}
\psi_i &\leftarrow \phi_i^{(1)} - \sum_{j=0}^{m-1} \langle \phi_i^{(1)}, \phi_j^{(0)} \rangle_{\mu_0} \phi_j^{(0)} - \sum_{l=0}^{i-1} \langle \phi_i^{(i)}, \psi_l \rangle_{\mu_0} \psi_l, \\
\psi_i &\leftarrow \frac{\psi_i}{||\psi_i||_{\mu_0}}.
\end{aligned} \tag{25}$$

The procedure in (25) results in a set of orthonormal basis of $W_0$ such that $\langle \psi_i, \psi_j \rangle_{\mu_0} = \delta_{ij}$ as well as $\langle \psi_i, \phi_j^{(0)} \rangle_{\mu_0}$, $\forall 0 \leq i < n, 0 \leq j < m$. We will see in Section C that the inner-product integrals in eq. (25) can be efficiently computed using the Gaussian Quadrature formulas (as discussed in Section B.2).

## C   Derivations for Multiwavelet Filters

Using the mathematical preliminaries and tools discussed in the Sections A and B, we are now in shape to present a detailed derivations for the measure dependent multiwavelet filters. We start with deriving the general filters expressions in Section C.1. Particular expressions for Legendre polynomials are presented in Section C.2, and then for Chebyshev polynomials in Section C.3.

### C.1   Filters as subspace projection coefficients

The 'multiwavelet filters' play the role of transforming the multiwavelet coefficients from one scale to another. Let us revisit the Section 2.2, where we defined a space of piecewise polynomial functions, for $k \in \mathbb{N}$ and $n \in \mathbb{Z}^+ \cup \{0\}$ as, $\mathbf{V}_n^k$. The $\dim(\mathbf{V}_n^k) = 2^n k$, and for subsequent $n$, each subspace is contained in another, i.e, $V_{n-1}^k \subset V_n^k$. Now, if $\phi_0, \ldots, \phi_{k-1}$ are a set of basis polynomials for $V_0^k$ w.r.t. measure $\mu_0$, then we know that a set of basis for $V_1^k$ can be obtained by scale and shift of $\phi_i$ as $\phi_{jl}^1 = 2^{1/2}\phi_j(2x - l)$ $l = 0, 1$, and the measure accordingly as $\mu_1$. For a given function $f$, its multiwavelet coefficients for projections over $V_0^k$ are taken as $s_{0i}^0 = \langle f, \phi_i \rangle_{\mu_0}$ and for $V_1^k$ is taken

as $s_{li}^1 = \langle f, \phi_{il}^1 \rangle_{\mu_1}$, and, we are looking for filter coefficients $(H)$ such that a transformation between projections at these two consecutive scale exists, or

$$s_{0i}^0 = \sum_{l=0,1} \sum_{j=0}^{k-1} H_{ij}^{(l)} s_{lj}^1. \tag{26}$$

Let us begin by considering a simple scenario. Since, $V_0^k \subset V_1^k$, the basis are related as

$$\phi_i = \sum_{j=0}^{k-1} \alpha_{ij}^{(0)} \sqrt{2}\phi_j(2x) + \sum_{j=0}^{k-1} \alpha_{ij}^{(1)} \sqrt{2}\phi_j(2x-1). \tag{27}$$

It is straightforward to see that if $\phi_i$ and $\phi_{il}^1$ are defined w.r.t. same measure, or $\mu_0 = \mu_1$ almost everywhere (a.e.), then the filters transforming the multiwavelet coefficients from higher to lower scale, are exactly equal to the subspace mapping coefficients $\alpha_{ij}^{(0)}, \alpha_{ij}^{(1)}$ ( by taking inner-product with $f$ on both sides in (27)). However, this is not the case in-general, i.e., the measures w.r.t. which the basis are defined at each scale are not necessarily same. To remedy this issue, and to generalize the multiwavelet filters, we now present a general measure-variant version of the multiwavelet filters.

We note that, solving for filters $H$ that satisfy eq. (26) indeed solves the general case of $n + 1 \rightarrow n$ scale, which can be obtained by a simple change of variables as $s_{l,i}^n = \sum_{j=0}^{k-1} H_{ij}^{(0)} s_{2l,j}^{n+1} + \sum_{j=0}^{k-1} H_{ij}^{(1)} s_{2l+1,j}^{n+1}$. Now, for solving (26), we consider the following equation

$$\phi_i(x)\frac{d\mu_0}{d\lambda}(x) = \sum_{j=0}^{k-1} H_{ij}^{(0)} \sqrt{2}\phi_j(2x)\frac{d\mu_1}{d\lambda}(x) + \sum_{j=0}^{k-1} H_{ij}^{(1)} \sqrt{2}\phi_j(2x-1)\frac{d\mu_1}{d\lambda}(x), \tag{28}$$

where $\frac{d\mu}{d\lambda}$ is the Radon-Nikodym derivative as discussed in Section B.1, and we have also defined $d\lambda := dx$. We observe that eq. (26) can be obtained from (28) by simply integrating with $f$ on both sides.

Next, we observe an important fact about multiwavelets (or wavelets in-general) that the advantages offered by multiwavelets rely on their ability to project a function *locally*. One way to achieve this is by computing basis functions which are dilation/translations of a fixed mother wavelet, for example, Figure 7. However, the idea can be generalized by projecting a given function onto any set of basis as long as they capture the *locality*. A possible approach to generalize is by using a tilt variant of the basis at higher scales, i.e., using $\sqrt{2}\tilde{\phi}_i(2x) = \sqrt{2}\phi_i(2x)\chi_0(x)$, and $\sqrt{2}\tilde{\phi}_i(2x-1) = \sqrt{2}\phi_i(2x-1)\chi_1(x)$ such that $\sqrt{2}\tilde{\phi}_i(2x)$ are now orthonormal w.r.t. weighting function $w(2x)/\chi_0^2(x)$, and similarly $\sqrt{2}\tilde{\phi}_i(2x-1)$ w.r.t. $w(2x-1)/\chi_1^2(x)$. By choosing $\chi_0(x) = w(2x)/w(x)$, and $\chi_1(x) = w(2x-1)/w(x)$, and taking the new tilted measure $\tilde{\mu}_1$ such that

$$\tilde{\mu}_1([0,1]) = \int_0^{1/2} \frac{w(2x)}{\chi_0^2(x)} d\lambda(x) + \int_{1/2}^1 \frac{w(2x-1)}{\chi_1^2(x)} d\lambda(x),$$

or,

$$\frac{d\tilde{\mu}_1}{d\lambda}(x) = \begin{cases} \frac{w(2x)}{\chi_0^2(x)} & 0 \le x \le 1/2, \\ \frac{w(2x-1)}{\chi_1^2(x)} & 1/2 < x \le 1. \end{cases}$$

We re-write the eq. (28), by substituting $\phi_i(2x) \leftarrow \tilde{\phi}_i(2x)$, $\phi_i(2x-1) \leftarrow \tilde{\phi}_i(2x-1)$ and $\mu_1 \leftarrow \tilde{\mu}_1$, in its most useful form for the current work as follows

$$\phi_i(x)w(x) = \sum_{j=0}^{k-1} H_{ij}^{(0)} \sqrt{2}\phi_j(2x)w(x) + \sum_{j=0}^{k-1} H_{ij}^{(1)} \sqrt{2}\phi_j(2x-1)w(x),$$

or,

$$\phi_i(x) = \sum_{j=0}^{k-1} H_{ij}^{(0)} \sqrt{2}\phi_j(2x) + \sum_{j=0}^{k-1} H_{ij}^{(1)} \sqrt{2}\phi_j(2x-1), \qquad (a.e.). \tag{29}$$

Thus, *filter coefficients can be looked upon as subspace projection coefficients*, with a proper choice of *tilted* basis. Note that eq.(33) is now equivalent to (27) but is an outcome of a different back-end machinery. Since, $\sqrt{2}\phi_i(2x), \sqrt{2}\phi_i(2x-1)$ are orthonormal basis for $V_1^k$, we have

$$2\int_0^{1/2} \phi_i(2x)\phi_j(2x)w(2x)dx = \delta_{ij}, \quad 2\int_{1/2}^1 \phi_i(2x-1)\phi_j(2x-1)w(2x-1)dx = \delta_{ij},$$

and hence we obtain the filter coefficients as follows

$$H_{ij}^{(0)} = \sqrt{2} \int_0^{1/2} \phi_i(x)\phi_j(2x)w(2x)dx, \tag{30}$$

$$H_{ij}^{(1)} = \sqrt{2} \int_{1/2}^1 \phi_i(x)\phi_j(2x-1)w(2x-1)dx. \tag{31}$$

For a given set of basis of $V_0^k$ as $\phi_0, \ldots, \phi_{k-1}$ defined w.r.t. measure/weight function $w(x)$, the filter coefficients $H$ can be derived by solving eq. (29). In a similar way, if $\psi_0, \ldots, \psi_{k-1}$ is the basis for the multiwavelet subspace $W_0^k$ w.r.t. measure $\mu_0$ such that $V_0^k \bigoplus W_0^k = V_1^k$, and the projection of function $f$ over $W_0^k$ is denoted by $d_{0,i}^0 = \langle f, \psi_i \rangle_{\mu_0}$, then the filter coefficients for obtaining the multiwavelet coefficients is written as

$$d_{0i}^0 = \sum_{l=0,1} \sum_{j=0}^{k-1} G_{ij}^{(l)} s_{lj}^1. \tag{32}$$

Again using a change of variables, we get $d_{l,i}^n = \sum_{j=0}^{k-1} G_{ij}^{(0)} s_{2l,j}^{n+1} + \sum_{j=0}^{k-1} G_{ij}^{(1)} s_{2l+1,j}^{n+1}$. To solve for $G$ in (32), similar to eq. (29), the measure-variant multiwavelet basis transformation (with appropriate tilt) is written as

$$\psi_i(x) = \sum_{j=0}^{k-1} G_{ij}^{(0)} \sqrt{2}\phi_j(2x) + \sum_{j=0}^{k-1} G_{ij}^{(1)} \sqrt{2}\phi_j(2x-1), \qquad (a.e.). \tag{33}$$

Similar to eq. (30)-(31), the filter coefficients $G$ can be obtained from (33) as follows

$$G_{ij}^{(0)} = \sqrt{2} \int_0^{1/2} \psi_i(x)\phi_j(2x)w(2x)dx, \tag{34}$$

$$G_{ij}^{(1)} = \sqrt{2} \int_{1/2}^1 \psi_i(x)\phi_j(2x-1)w(2x-1)dx. \tag{35}$$

Since $\langle \phi_i, \phi_j \rangle_{\mu_0} = \delta_{ij}$, $\langle \psi_i, \psi_j \rangle_{\mu_0} = \delta_{ij}$ and $\langle \phi_i, \psi_j \rangle_{\mu_0} = 0$, therefore, using (29), (33), we can write that

$$\int_0^1 \phi_i(x)\phi_j(x)w(x)dx = 2\sum_{l=0}^{k-1}\sum_{l'=0}^{k-1} H_{il}^{(0)} H_{jl'}^{(0)} \int_0^{1/2} \phi_l(2x)\phi_{l'}(2x)w(x)dx$$
$$+ 2\sum_{l=0}^{k-1}\sum_{l'=0}^{k-1} H_{il}^{(1)} H_{jl'}^{(1)} \int_{1/2}^1 \phi_l(2x-1)\phi_{l'}(2x-1)w(x)dx, \tag{36}$$

$$\int_0^1 \psi_i(x)\psi_j(x)w(x)dx = 2\sum_{l=0}^{k-1}\sum_{l'=0}^{k-1} G_{il}^{(0)} G_{jl'}^{(0)} \int_0^{1/2} \phi_l(2x)\phi_{l'}(2x)w(x)dx$$
$$+ 2\sum_{l=0}^{k-1}\sum_{l'=0}^{k-1} G_{il}^{(1)} G_{jl'}^{(1)} \int_{1/2}^1 \phi_l(2x-1)\phi_{l'}(2x-1)w(x)dx, \tag{37}$$

$$0 = 2\sum_{l=0}^{k-1}\sum_{l'=0}^{k-1} H_{il}^{(0)} G_{jl'}^{(0)} \int_0^{1/2} \phi_l(2x)\phi_{l'}(2x)w(x)dx$$
$$+ 2\sum_{l=0}^{k-1}\sum_{l'=0}^{k-1} H_{il}^{(1)} G_{jl'}^{(1)} \int_{1/2}^1 \phi_l(2x-1)\phi_{l'}(2x-1)w(x)dx. \tag{38}$$

Let us define filter matrices as $H^{(l)} = [H_{ij}^{(l)}] \in \mathbb{R}^{k \times k}$ and $G^{(l)} = [G_{ij}^{(l)}] \in \mathbb{R}^{k \times k}$ for $l = 0, 1$. Also, we define correction matrices as $\Sigma^{(0)} = [\Sigma_{ij}^{(0)}], \Sigma^{(1)} = [\Sigma_{ij}^{(1)}]$ such that

$$\Sigma_{ij}^{(0)} = 2 \int_0^{1/2} \phi_i(2x)\phi_j(2x)w(x)dx,$$
$$\Sigma_{ij}^{(1)} = 2 \int_{1/2}^1 \phi_i(2x-1)\phi_j(2x-1)w(x)dx. \tag{39}$$

Now, we can write that

$$H^{(0)}\Sigma^{(0)}H^{(0)\,T} + H^{(1)}\Sigma^{(1)}H^{(1)\,T} = I,$$
$$G^{(0)}\Sigma^{(0)}G^{(0)\,T} + G^{(1)}\Sigma^{(1)}G^{(1)\,T} = I, \tag{40}$$
$$H^{(0)}\Sigma^{(0)}G^{(0)\,T} + H^{(1)}\Sigma^{(1)}G^{(1)\,T} = 0.$$

Rearranging eq. we can finally express the relationships between filter matrices and correction matrices as follows

$$\begin{bmatrix} H^{(0)} & H^{(1)} \\ G^{(0)} & G^{(1)} \end{bmatrix} \begin{bmatrix} \Sigma^{(0)} & 0 \\ 0 & \Sigma^{(1)} \end{bmatrix} \begin{bmatrix} H^{(0)} & H^{(1)} \\ G^{(0)} & G^{(1)} \end{bmatrix}^{T} = I. \tag{41}$$

The discussion till now is related to 'decomposition' or transformation of multiwavelet transform coefficients from higher to lower scale. However, the other direction, i.e., 'reconstruction' or transformation from lower to higher scale can also be obtained from (41). First, note that the general form of eq. (26), (32) can be written in the matrix format as

$$\mathbf{s}_l^n = H^{(0)}\mathbf{s}_{2l}^{n+1} + H^{(1)}\mathbf{s}_{2l+1}^{n+1},$$
$$\mathbf{d}_l^n = G^{(0)}\mathbf{s}_{2l}^{n+1} + G^{(1)}\mathbf{s}_{2l+1}^{n+1}. \tag{42}$$

Next, we observe that $\Sigma^{(0)}, \Sigma^{(1)} \succ 0$, which follows from their definition. Therefore, eq. (41) can be inverted to get the following form

$$\begin{bmatrix} H^{(0)} & H^{(1)} \\ G^{(0)} & G^{(1)} \end{bmatrix} \begin{bmatrix} H^{(0)} & H^{(1)} \\ G^{(0)} & G^{(1)} \end{bmatrix}^{T} = \begin{bmatrix} \Sigma^{(0)\,-1} & 0 \\ 0 & \Sigma^{(1)\,-1} \end{bmatrix}. \tag{43}$$

Finally, by using (43), we can essentially invert the eq. (42) to get

$$\mathbf{s}_{2l}^{n+1} = \Sigma^{(0)}(H^{(0)\,T}\mathbf{s}_l^n + G^{(0)\,T}\mathbf{d}_l^n),$$
$$\mathbf{s}_{2l+1}^{n+1} = \Sigma^{(1)}(H^{(1)\,T}\mathbf{s}_l^n + G^{(1)\,T}\mathbf{d}_l^n). \tag{44}$$

In the following Section C.2, C.3 we see the the filters $H, G$ in (42), (44) for different polynomial basis.

## C.2 Multiwavelets using Legendre Polynomials

The basis for $V_0^k$ are chosen as normalized shifted Legendre polynomials of degree upto $k$ w.r.t. weight function $w_L(2x-1) = \mathbf{1}_{[0,1]}(x)$ from Section A.2.1. For example, the first three bases are

$$\phi_0(x) = 1,$$
$$\phi_1(x) = \sqrt{3}(2x - 1), \tag{45}$$
$$\phi_2(x) = \sqrt{5}(6x^2 - 6x + 1), \quad 0 \le x \le 1.$$

For deriving a set of basis $\psi_i$ of $W_0^k$ using GSO, we need to evaluate the integrals which could be done efficiently using Gaussian quadrature.

**Gaussian-Legendre Quadrature:** The integrals involved in GSO procedure, and the computations of $H, G$ can be done efficiently using the Gaussian quadrature as discussed in Section B.2. Since the basis functions $\phi_i, \psi_i$ are polynomials, therefore, the quadrature summation would be *exact*. For a given $k$ basis of the subspace $V_0^k$, the deg $(\phi_i \phi_j) < 2k - 1$, as well as deg $(\phi_i \psi_j) < 2k - 1$, therefore a $k$-point quadrature would be sufficient for expressing the integrals. Next, we take the interval $[a, b] = [0, 1]$, and the OPs for approximation in Gaussian quadrature as shifted Legendre polynomials $P_k(2x - 1)$. The weight coefficients $\omega_i$ can be written as

$$\omega_i = \frac{a_k}{a_{k-1}} \frac{\int_0^1 P_{k-1}^2(2x-1)w(2x-1)dx}{P_k'(2x_i-1)P_{k-1}(2x_i-1)}$$
$$= \frac{2k-1}{k} \cdot \frac{1}{2k-1} \frac{1}{P_k'(2x_i-1)P_{k-1}(2x_i-1)} = \frac{1}{kP_k'(2x_i-1)P_{k-1}(2x_i-1)}, \tag{46}$$

where $x_i$ are the $k$ roots of $P_k(2x-1)$ and $a_k$ can be expressed in terms of $a_{k-1}$ using the recurrence relationship of Legendre polynomials from Section A.2.1.

A set of basis for $V_1^k$ is $\sqrt{2}\phi_i(2x)$ and $\sqrt{2}\phi_i(2x-1)$ with weight functions $w_L(4x-1) = \mathbf{1}_{[0,1/2]}(x)$ and $w_L(4x-3) = \mathbf{1}_{(1/2,1]}(x)$, respectively. We now use GSO procedure as outlined in Section B.3 to obtain set of basis $\psi_0, \ldots, \psi_{k-1}$ for $W_0^k$. We use Gaussian-Legendre quadrature formulas for computing the inner-products. As an example, the inner-products are computed as follows

$$\langle \sqrt{2}\phi_i, \phi_j \rangle_{\mu_0} = \int_0^1 \sqrt{2}\phi_i(2x)\phi_j(x)w_L(2x-1)dx$$

$$= \sqrt{2}\sum_{i=1}^k \omega_i \phi_i(2x_i)\phi_j(x_i),$$

where $\phi_i(2x_i) = 0$ for $x_i > 0.5$.

With shifted Legendre polynomials as basis for $V_0^3$, the multiwavelet bases for $W_0^3$ are

$$\psi_0(x) = \begin{cases} 6x - 1 & 0 \le x \le 1/2, \\ 6x - 5 & 1/2 < x \le 1, \end{cases}$$

$$\psi_1(x) = \begin{cases} \sqrt{3}(30x^2 - 14x + 1) & 0 \le x \le 1/2, \\ \sqrt{3}(30x^2 - 46x + 17) & 1/2 < x \le 1, \end{cases} \tag{47}$$

$$\psi_2(x) = \begin{cases} \sqrt{5}(24x^2 - 12x + 1) & 0 \le x \le 1/2, \\ \sqrt{5}(-24x^2 + 36x - 13) & 1/2 < x \le 1. \end{cases}$$

Next, we compute the filter matrices, but first note that since the weighting function for Legendre polynomials basis are $w_L(x) = \mathbf{1}_{[0,1]}(x)$, therefore, $\Sigma^{(0)}, \Sigma^{(1)}$ in eq. (39) are just identity matrices because of orthonormality of the basis $\sqrt{2}\phi_i(2x)$ and $\sqrt{2}\phi_i(2x-1)$ w.r.t. $\mathbf{1}_{[0,1/2]}(x)$ and $\mathbf{1}_{[1/2,1]}(x)$, respectively. The filter coefficients can be computed using Gaussian-Legendre quadrature as follows

$$H_{ij}^{(0)} = \sqrt{2}\int_0^{1/2} \phi_i(x)\phi_j(2x)w_L(2x-1)dx$$

$$= \frac{1}{\sqrt{2}}\int_0^1 \phi_i(x/2)\phi_j(x)dx$$

$$= \frac{1}{\sqrt{2}}\sum_{i=1}^k \omega_i \phi_i\left(\frac{x_i}{2}\right)\phi_j(x_i),$$

and similarly other coefficients can be obtained in eq. (30)-(31), (34)-(35). As an example, for $k = 3$, following the outlined procedure, the filter coefficients are derived as follows

$$H^{(0)} = \begin{bmatrix} \frac{1}{\sqrt{2}} & 0 & 0 \\ -\frac{\sqrt{3}}{2\sqrt{2}} & \frac{1}{2\sqrt{2}} & 0 \\ 0 & -\frac{\sqrt{15}}{4\sqrt{2}} & \frac{1}{4\sqrt{2}} \end{bmatrix}, \qquad H^{(1)} = \begin{bmatrix} \frac{1}{\sqrt{2}} & 0 & 0 \\ \frac{\sqrt{3}}{2\sqrt{2}} & \frac{1}{2\sqrt{2}} & 0 \\ 0 & \frac{\sqrt{15}}{4\sqrt{2}} & \frac{1}{4\sqrt{2}} \end{bmatrix},$$

$$G^{(0)} = \begin{bmatrix} \frac{1}{2\sqrt{2}} & \frac{\sqrt{3}}{2\sqrt{2}} & 0 \\ 0 & \frac{1}{4\sqrt{2}} & \frac{\sqrt{15}}{4\sqrt{2}} \\ 0 & 0 & \frac{1}{\sqrt{2}} \end{bmatrix}, \qquad G^{(1)} = \begin{bmatrix} -\frac{1}{2\sqrt{2}} & \frac{\sqrt{3}}{2\sqrt{2}} & 0 \\ 0 & -\frac{1}{4\sqrt{2}} & \frac{\sqrt{15}}{4\sqrt{2}} \\ 0 & 0 & -\frac{1}{\sqrt{2}} \end{bmatrix}.$$

### C.3 Multiwavelets using Chebyshev Polynomials

We choose the basis for $V_0^k$ as shifted Chebyshev polynomials of the first-order from degree $0$ to $k-1$. The weighting function for shifted Chebyshev polynomials is $w_{Ch}(2x-1) = 1\sqrt{1-(2x-1)^2}$

from Section A.2.2. The first three bases using Chebyshev polynomials are as follows

$$\phi_0(x) = \sqrt{2/\pi},$$

$$\phi_1(x) = \frac{2}{\sqrt{\pi}}(2x - 1),$$

$$\phi_2(x) = \frac{2}{\sqrt{\pi}}(8x^2 - 8x + 1), 0 \leq x \leq 1.$$

The Gaussian quadrature for the Chebyshev polynomials is used to evaluate the integrals that appears in the GSO procedure as well as in the computations of filters $H, G$.

**Gaussian-Chebyshev Quadrature:** The basis functions $\phi_i, \psi_i$ resulting from the use of shifted Chebyshev polynomials are also polynomials with degree of their products such that $\deg(\phi_i\phi_j) < 2k - 1$ and $\deg(\phi_i\psi_i) < 2k - 1$, therefore a $k$-point quadrature would be sufficient for evaluating the integrals that have products of bases. Upon taking the interval $[a, b]$ as $[0, 1]$, and using the canonical OPs as shifted Chebyshev polynomials, the weight coefficients are written as

$$\begin{aligned}
\omega_i &= \frac{a_k}{a_{k-1}} \frac{\int_0^1 T_{k-1}^2(2x - 1)w_{Ch}(2x - 1)dx}{T_k'(2x_i - 1)T_{k-1}(2x_i - 1)} \\
&\overset{(a)}{=} 2\frac{\pi}{4} \frac{1}{T_k'(2x_i - 1)T_{k-1}(2x_i - 1)} \\
&\overset{(b)}{=} \frac{\pi}{2k},
\end{aligned} \tag{48}$$

where $x_i$ are the $k$ roots of $T_k(2x - 1)$, $(a)$ is using the fact that $a_n/a_{n-1} = 2$ by using the recurrence relationship of Chebyshev polynomials from Section A.2.2, and assumes $k > 1$ for the squared integral. For $(b)$, we first note that $T_k(\cos\theta) = \cos(k\theta)$, hence, $T_k'(\cos\theta) = n\sin(n\theta)/\sin(\theta)$. Since $x_i$ are the roots of $T_k(2x - 1)$, therefore, $2x_i - 1 = \cos(\frac{\pi}{n}(i - 1/2))$. Substituting the $x_i$, we get $T_k'(2x_i - 1)T_{k-1}(2x_i - 1) = k$.

A set of basis for $V_1^k$ is $\sqrt{2}\phi_i(2x)$ and $\sqrt{2}\phi_i(2x - 1)$ with weight functions $w_{Ch}(4x - 1) = 1/\sqrt{1 - (4x - 1)^2}$ and $w_{Cb}(4x - 3) = 1/\sqrt{1 - (4x - 3)^2}$, respectively. We now use GSO procedure as outlined in Section B.3 to obtain set of basis $\psi_0, \ldots, \psi_{k-1}$ for $W_0^k$. We use Gaussian-Chebyshev quadrature formulas for computing the inner-products. As an example, the inner-products are computed as follows

$$\begin{aligned}
\langle \sqrt{2}\phi_i, \phi_j \rangle_{\mu_0} &= \int_0^1 \sqrt{2}\phi_i(2x)\phi_j(x)w_{Ch}(2x - 1)dx \\
&= \sqrt{2}\frac{\pi}{2k} \sum_{i=1}^k \phi_i(2x_i)\phi_j(x_i),
\end{aligned}$$

where $\phi_i(2x_i) = 0$ for $x_i > 0.5$.

With shifted Chebyshev polynomials as basis for $V_0^3$, the multiwavelet bases for $W_0^3$ are derived as

$$\begin{aligned}
\psi_0(x) &= \begin{cases} 4.9749x - 0.5560 & 0 \leq x \leq 1/2, \\ 4.9749x - 4.4189 & 1/2 < x \leq 1, \end{cases} \\
\psi_1(x) &= \begin{cases} 58.3516x^2 - 22.6187x + 0.9326 & 0 \leq x \leq 1/2, \\ 58.3516x^2 - 94.0846x + 36.6655 & 1/2 < x \leq 1, \end{cases} \\
\psi_2(x) &= \begin{cases} 59.0457x^2 - 23.7328x + 1.0941 & 0 \leq x \leq 1/2, \\ -59.0457x^2 + 94.3586x - 36.4070 & 1/2 < x \leq 1. \end{cases}
\end{aligned} \tag{49}$$

Next, we compute the filter and the correction matrices. The filter coefficients can be computed using Gaussian-Chebyshev quadrature as follows

$$
\begin{aligned}
H_{ij}^{(0)} &= \sqrt{2} \int_0^{1/2} \phi_i(x)\phi_j(2x)w_{Ch}(4x-1)dx \\
&= \frac{1}{\sqrt{2}} \int_0^1 \phi_i(x/2)\phi_j(x)w_{Ch}(2x-1)dx \\
&= \frac{\pi}{2\sqrt{2}k} \sum_{i=1}^k \phi_i\left(\frac{x_i}{2}\right)\phi_j(x_i),
\end{aligned}
$$

and similarly, other coefficients can be obtained in eq. (30)-(31), (34)-(35). Using the outlined procedure for Chebyshev based OP basis, for $k=3$, the filter and the corrections matrices are derived as

$$
H^{(0)} = \begin{bmatrix} \frac{1}{\sqrt{2}} & 0 & 0 \\ -\frac{1}{2} & \frac{1}{2\sqrt{2}} & 0 \\ -\frac{1}{4} & -\frac{1}{\sqrt{2}} & \frac{1}{4\sqrt{2}} \end{bmatrix}, \qquad\qquad H^{(1)} = \begin{bmatrix} \frac{1}{\sqrt{2}} & 0 & 0 \\ \frac{1}{2} & \frac{1}{2\sqrt{2}} & 0 \\ -\frac{1}{4} & \frac{1}{\sqrt{2}} & \frac{1}{4\sqrt{2}} \end{bmatrix},
$$

$$
G^{(0)} = \begin{bmatrix} 0.6094 & 0.7794 & 0 \\ 0.6632 & 1.0272 & 1.1427 \\ 0.6172 & 0.9070 & 1.1562 \end{bmatrix}, \qquad G^{(1)} = \begin{bmatrix} -0.6094 & 0.7794 & 0 \\ 0.6632 & -1.0272 & 1.1427 \\ -0.6172 & 0.9070 & -1.1562 \end{bmatrix},
$$

$$
\Sigma^{(0)} = \begin{bmatrix} 1 & -0.4071 & -0.2144 \\ -0.4071 & 0.8483 & -0.4482 \\ -0.2144 & -0.4482 & 0.8400 \end{bmatrix}, \qquad \Sigma^{(1)} = \begin{bmatrix} 1 & 0.4071 & -0.2144 \\ 0.4071 & 0.8483 & 0.4482 \\ -0.2144 & 0.4482 & 0.8400 \end{bmatrix}.
$$

### C.4 Numerical Considerations

The numerical computations of the filter matrices are done using Gaussian quadrature as discussed in Sections C.2 and C.3 for Legendre and Chebyshev polynomials, respectively. For odd $k$, a root of the canonical polynomial (either Legendre, Chebyshev) would be exactly $0.5$. Since the multiwavelets bases $\psi_i$ for $W_0^k$ are discontinuous at $0.5$, the quadrature sum can lead to an unexpected result due to the finite-precision of the roots $x_i$. One solution for this is to add a small number $\epsilon, \epsilon > 0$ to $x_i$ to avoid the singularity. Another solution, which we have used, is to perform a $\tilde{k}$-quadrature, where $\tilde{k} = 2k$. Note that, any high value of quadrature sum would work as long as it is greater than $k$, and we choose an even value to avoid the root at the singularity ($x = 0.5$).

To check the validity of the numerically computed filter coefficients from the Gaussian quadrature, we can use eq. (41). In a $k$-point quadrature, the summation involves up to $k$ degree polynomials, and we found that for large values of $k$, for example, $k > 20$, the filter matrices tend to diverge from the mathematical constraint of (41). Note that this is not due to the involved mathematics but the precision offered by floating-point values. For the current work, we found values of $k$ in the range of $[1, 6]$ to be most useful. However, a future research should look into other possible alternatives to work around the numerical errors due to the floating-point precision.

## D   Additional Results

We present numerical evaluation of the proposed multiwavelets-based models on an additional dataset of Navier-Stokes in Section D.1. Next, in Section D.2, we present numerical results for prediction at finer resolutions with the use of lower-resolution trained models. Section D.3 presents additional results on the evaluation of multiwavelets on pseudo-differential equations.

### D.1   Navier-Stokes Equation

Navier-Stokes Equations [2, 10] describe the motion of viscous fluid substances, which can be used to model the ocean currents, the weather, and air flow. We experiment on the 2-d Navier-Stokes equation for a viscous, incompressible fluid in vorticity form on the unit torus, where it takes the

| Networks | $\nu = 1e-3$ $T = 50$ $N = 1000$ | $\nu = 1e-4$ $T = 30$ $N = 1000$ | $\nu = 1e-4$ $T = 30$ $N = 10000$ | $\nu = 1e-5$ $T = 20$ $N = 1000$ |
|---|---|---|---|---|
| MWT Leg | **0.00625** | **0.1518** | **0.0667** | **0.1541** |
| MWT Chb | 0.00720 | 0.1574 | 0.0720 | 0.1667 |
| FNO-3D | 0.0086 | 0.1918 | 0.0820 | 0.1893 |
| FNO-2D | 0.0128 | 0.1559 | 0.0973 | 0.1556 |
| U-Net | 0.0245 | 0.2051 | 0.1190 | 0.1982 |
| TF-Net | 0.0225 | 0.2253 | 0.1168 | 0.2268 |
| Res-Net | 0.0701 | 0.2871 | 0.2311 | 0.2753 |

Table 3: Navier-Stokes Equation validation at various viscosities $\nu$. Top: Our methods. Bottom: previous works of Neural operators and other deep learning models.

following form:

$$\frac{\partial w(x,t)}{\partial t} + u(x,t) \cdot \nabla w(x,t) - \nu \Delta w(x,t) = f(x), \quad x \in (0,1)^2, t \in (0,T]$$
$$\nabla \cdot u(x,t) = 0, \qquad\qquad x \in (0,1)^2, t \in [0,T] \qquad (50)$$
$$w_0(x) = w(x, t=0), \qquad\qquad x \in (0,1)^2$$

We set the experiments to learn the operator mapping the vorticity $w$ up to time 10 to $w$ at a later time $T > 10$. More specifically, task for the neural operator is to map the first $T$ time units to last $T - 10$ time units of vorticity $w$. To compare with the state-of-the-art model FNO [49] and other configurations under the same conditions, we use the same Navier-Stokes' data and the results that have been published in [49]. The initial condition is sampled as Gaussian random fields where $w_0 \sim \mathcal{N}(0, 7^{\frac{3}{2}}(-\Delta + 7^2 I)^{-2.5})$ with periodic boundary conditions. The forcing function $f(x) = 0.1(\sin(2\pi(x_1 + x_2)) + \cos(2\pi(x_1 + x_2)))$. The experiments are conducted with ① the viscosities $\nu = 1e - 3$, the final time $T = 50$, the number of training pairs $N = 1000$; ② $\nu = 1e-4, T = 30, N = 1000$; ③ $\nu = 1e-4, T = 30, N = 10000$; ④ $\nu = 1e-5, T = 20, N = 1000$. The data sets are generated on a $256 \times 256$ grid and are subsampled to $64 \times 64$.

We see in Table 3 that the proposed MWT Leg outperforms the existing Neural operators as well as other deep NN benchmarks. The MWT models have used a 2d multiwavelet transform with $k = 3$ for the vorticity $w$, and 3d convolutions in the $A, B, C$ NNs for estimating the time-correlated kernels. The MWT models (both Leg and Chb) are trained for 500 epochs for all the experiments except for $N = 10000, T = 30, \nu = 1e - 4$ case where the models are trained for 200 epochs. Note that similar to FNO-2D, a time-recurrent version of the MWT models could also be trained and most likely will improve the resulting $L2$ error for the less data setups like $N = 1000, \nu = 1e - 4$ and $N = 1000, \nu = 1e - 5$. However, in this work we have only experimented with the 3d convolutions (for $A, B, C$) version.

### D.2 Prediction at higher resolutions

The proposed multiwavelets-based operator learning model is resolution-invariant by design. Upon learning an operator map between the function spaces, the proposed models have the ability to generalize beyond the training resolution. In this Section, we evaluate the resolution extension property of the MWT models using the Burgers' equation dataset as described in the Section 3.3. A pipeline for the experiment is shown in Figure 8. The numerical results for the experiments are shown in Table 4. We see that on training with a lower resolution, for example, $s = 256$, the prediction

| Train \ Test | s = 2048 | s = 4096 | s = 8192 |
|---|---|---|---|
| s=128 | 0.0368 | 0.0389 | 0.0456 |
| s=256 | 0.0226 | 0.0281 | 0.0321 |
| s=512 | 0.0140 | 0.0191 | 0.0241 |

Table 4: MWT Leg model trained at lower resolutions can predict the output at higher resolutions.

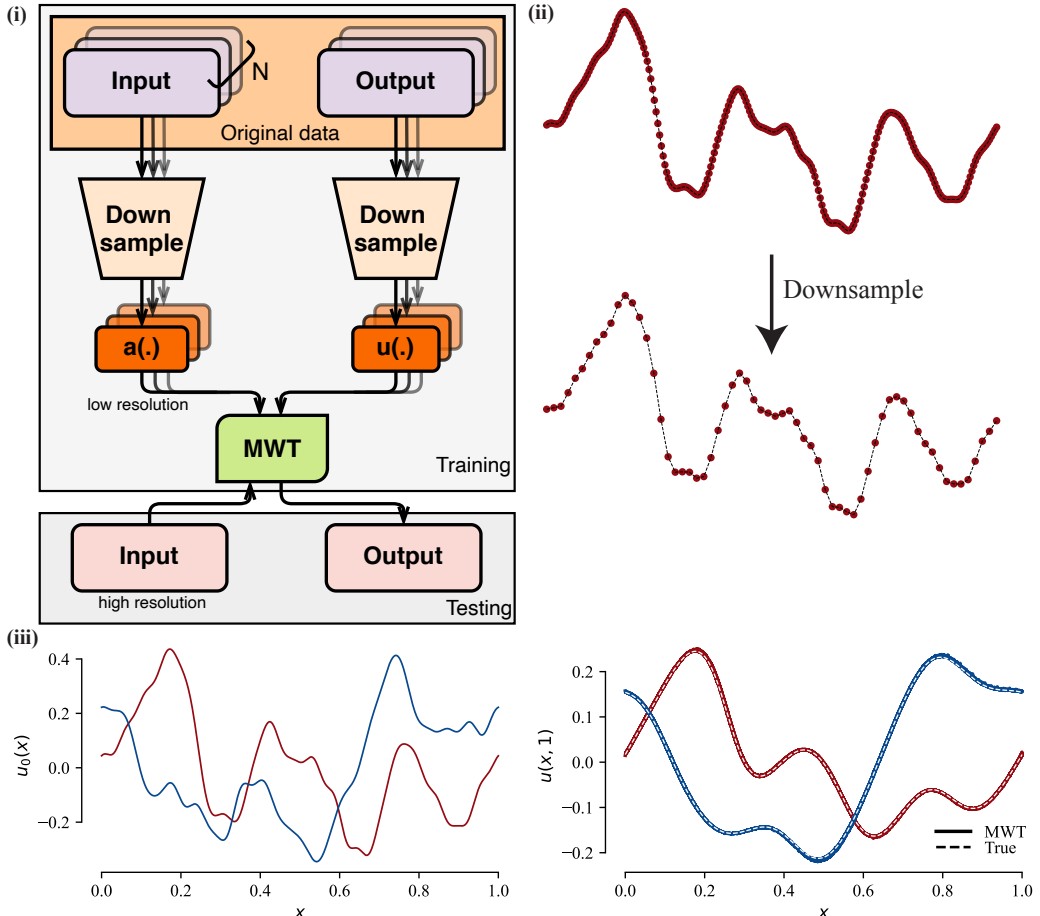

Figure 8: **Prediction at higher resolution:** The proposed model (MWT) learns the function mapping using the data with a coarse resolution, and can predict the output at a higher resolution. **(i)** The resolution-extension experiment pipeline. **(ii)** An example of down-sampling of the associated functions used in the training. **(iii)** We show two test samples with example-1 marked as blue while example-2 is marked as red. **Left:** input functions ($u_0$) of the examples. **Right:** corresponding outputs $u(x, 1)$ at $s = 8192$ from MWT Leg (trained on $s = 256$) of the 2 examples, and their higher-resolution ($s = 8192$) ground truth (dotted line).

error at 10X higher resolution $s = 2048$ is 0.0226, or 2.26%. A sample input/output for learning at $s = 256$ while predicting a $s = 8192$ resolution is shown in Figure 8. Also, learning at an even coarser resolution of $s = 128$, the proposed model can predict the output of $2^6$ times the resolution (i.e., $s = 8192$) data with an relative $L2$ error of 4.56%.

### D.3 Pseudo-Differential Equation

Similar to the experiments presented in the Section 3.2 for Euler-Bernoulli equation, we now present an additional result on a different pseudo-differential equation. We modify the Euler-Bernoulli beam to a 3rd order PDE as follows

$$\frac{\partial^3 u}{\partial x^3} - \omega^2 u = f(x), \quad \left.\frac{\partial u}{\partial x}\right|_{x=0} = 0, \quad x \in [0, 1]$$
$$u(0) = u(1) = 0, \tag{51}$$

where $u(x)$ is the Fourier transform of the time-varying displacement, $\omega = 215$ is the frequency, $f(x)$ is the external function. The eq. (51) is not known to have a physical meaning like Euler-Bernoulli, however, from a simulation point-of-view it is sufficient to be used as a canonical PDE. A sample force function (input) and

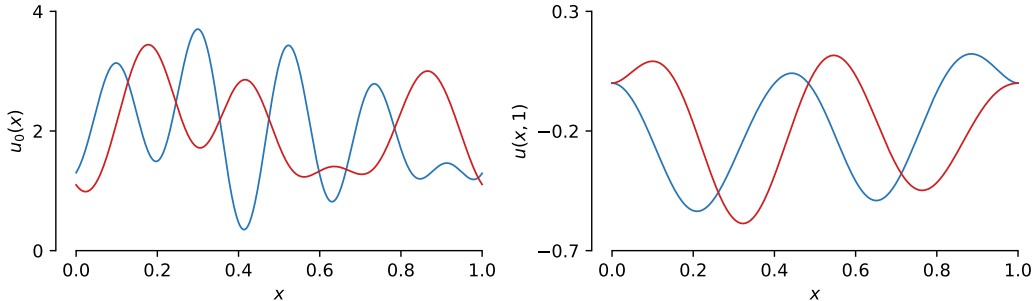

Figure 10: Two examples of 4th order Euler-Bernoulli equation. **Left:** Two input functions ($u_0$) in different colors. **Right:** corresponding outputs ($u(x, 1)$) in the same color.

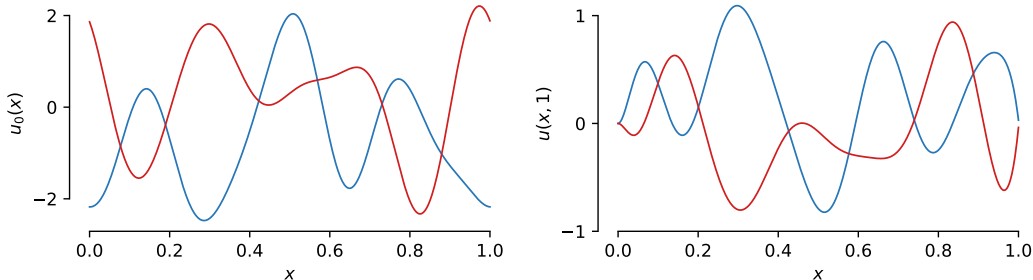

Figure 11: Sample input/output for the PDE of Section D.3. **Left:** Two input functions ($u_0$) examples in Red and Blue. **Right:** corresponding outputs ($u(x, 1)$) in the same color.

the solution of the PDE in (51) is shown in Figure 11. The eq. (51) is a pseudo-differential equation with the maximum derivative order $T + 1 = 3$. We now take the task of learning the map from $f$ to $u$. In Figure 9, we see that for $k \geq 2$, the models relative error across epochs is similar, which again is in accordance with the Property 1, i.e., $k > T - 1$ is sufficient for annihilation the kernel away from the diagonal by multiwavelets. We saw a similar pattern for the 4th order PDE in Section 3.2 but for $k \geq 3$.

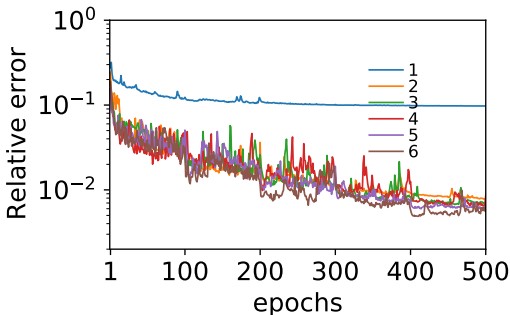

Figure 9: Relative $L2$ error vs epochs for MWT Leg with different number of OP basis $k$.

### D.4 Korteweg-de Vries (KdV) Equation

We present additional results for the KdV equation (see Section 3.1). First, we demonstrate the operator learning when the input is sampled from a squared exponential kernel. Second, we experiment on the learning behavior of the Neural operators when the train and test samples are generated from different random sampling schemes.

#### D.4.1 Squared Exponential Kernel

We sample the input $u_0(x)$ from a squared exponential kernel, and solve the KdV equation in a similar setting as mentioned in Section 3.1. Due to the periodic boundary conditions, a periodic version of the squared exponential kernel [60] is used as follows.

$$k(x, x') = \exp\left(-2\frac{\sin^2(\pi(x - x')/P)}{L^2}\right),$$

where, $P$ is the domain length and $L$ is the smoothing parameter of the kernel. The random input function is sampled from $\mathcal{N}(0, K_m)$ with $K_m$ being the kernel matrix by taking $P = 1$ (domain

| Networks | s = 64 | s = 128 | s = 256 | s = 512 | s = 1024 |
|---|---|---|---|---|---|
| MWT Leg | **0.00190** | **0.00214** | **0.00204** | **0.00201** | **0.00211** |
| MWT Chb | 0.00239 | 0.00241 | 0.00221 | 0.00289 | 0.00348 |
| FNO | 0.00335 | 0.00330 | 0.00375 | 0.00402 | 0.00456 |
| MGNO | 0.0761 | 0.0593 | 0.0724 | 0.0940 | 0.0660 |
| LNO | 0.0759 | 0.0756 | 0.0673 | 0.0807 | 0.0792 |
| GNO | 0.0871 | 0.0801 | 0.0722 | 0.0798 | 0.0777 |

Table 5: Korteweg-de Vries (KdV) equation benchmarks for different input resolution $s$ with input $u_0(x)$ sampled from a squared exponential kernel. Top: Our methods. Bottom: previous works of Neural operator.

| Networks | $\lambda = 0.25$ | $\lambda = 0.5$ | $\lambda = 0.75$ |
|---|---|---|---|
| | | $s = 64$ | |
| MWT Leg | 0.00819 | 0.00413 | **0.00202** |
| MWT Chb | **0.00751** | **0.00347** | 0.00210 |
| FNO | 0.0141 | 0.00822 | 0.00404 |
| MGNO | 0.3701 | 0.2030 | 0.0862 |
| LNO | 0.1012 | 0.0783 | 0.0141 |
| | | $s = 256$ | |
| MWT Leg | 0.00690 | **0.00322** | **0.00145** |
| MWT Chb | **0.00616** | 0.00344 | 0.00182 |
| FNO | 0.0134 | 0.00901 | 0.00376 |
| MGNO | 0.4492 | 0.2114 | 0.1221 |
| LNO | 0.1306 | 0.0821 | 0.0161 |
| | | $s = 1024$ | |
| MWT Leg | **0.00641** | 0.00408 | **0.00127** |
| MWT Chb | 0.00687 | **0.00333** | 0.00176 |
| FNO | 0.0141 | 0.00718 | 0.00359 |
| MGNO | 0.4774 | 0.2805 | 0.1309 |
| LNO | 0.1140 | 0.0752 | 0.0139 |

Table 6: Neural operators performance when training on random inputs sampled from Squared exponential kernel and testing on samples generated from smooth random functions [32] with controllable parameter $\lambda$. The random functions are used as the input $u_0(x)$ for Korteweg-de Vries (KdV) equation as mentioned in Section 3.1. In the test data, $\lambda$ is inversely proportional to sharpness of the fluctuations.

length) and $L = 0.5$ to avoid the sharp peaks in the sampled function. The results for the Neural operators (similar to Table 1) is shown in Table 5. We see that MWT models perform better than the existing neural operators at all resolutions.

### D.4.2 Training/Evaluation with different sampling rules

The experiments in the current work and also in all of the recent neural operators work [49, 51] have used the datasets such that the train and test samples are generated by sampling the input function using the same rule. For example, in KdV, a complete dataset is first generated by randomly sampling the inputs $u_0(x)$ from $\mathcal{N}(0, 7^4(\Delta + 7^2 I)^{-2.5})$ and then splitting the dataset into train/test. This setting is useful when dealing with the systems such that the future evaluation function samples have similar patterns like smoothness, periodicity, presence of peaks. However, from the viewpoint of learning the operator between the function spaces, this is not a general setting. We have seen in Figure 4 that upon varying the fluctuation strength in the inputs (both train and test), the performance of the neural operators differ. We now perform an addition experiment in which the neural operator is trained using the samples from a periodic squared exponential kernel (Section D.4.1) and evaluated on the samples generated from random fields [32] with fluctuation parameter $\lambda$. We see in Table 6 that instead of different generating rules, the properties like fluctuation strength matters more when it comes to learning the operator map. Evaluation on samples that are generated from a different rule can still work well provided that the fluctuations are of similar nature. It is intuitive that by learning only from the low-frequency signals, the generalization to higher-frequency signals is difficult.

| Networks | s = 256 | s = 512 | s = 1024 | s = 2048 | s = 4096 | s = 8192 |
|---|---|---|---|---|---|---|
| MWT Leg | **0.00199** | **0.00185** | **0.00184** | **0.00186** | **0.00185** | **0.00178** |
| MWT Chb | 0.00402 | 0.00381 | 0.00336 | 0.00395 | 0.00299 | 0.00289 |
| FNO | 0.00332 | 0.00333 | 0.00377 | 0.00346 | 0.00324 | 0.00336 |
| MGNO | 0.0243 | 0.0355 | 0.0374 | 0.0360 | 0.0364 | 0.0364 |
| LNO | 0.0212 | 0.0221 | 0.0217 | 0.0219 | 0.0200 | 0.0189 |
| GNO | 0.0555 | 0.0594 | 0.0651 | 0.0663 | 0.0666 | 0.0699 |

Table 7: Burgers' Equation validation at various input resolution $s$. Top: Our methods. Bottom: previous works of Neural operators.

## D.5 Burgers Equation

The numerical values for the Burgers' equation experiment, as presented in Figure 6, is provided in the Table 7.