# OpenReview forum: "Multiwavelet-based Operator Learning for Differential Equations"
_NeurIPS.cc/2021/Conference — NeurIPS 2021 Spotlight_

### Official Review · Reviewer_aK5Q · 2021-07-10

**Rating:** 8
**Confidence:** 4

**Summary:**

The manuscript entitled “Multiwavelet-based Operator Learning for Differential Equations” exploits the fundamental properties of the operator’s kernel which enable numerically efficient representation. The proposed method exhibits a 10X improvement on relative L2 error for some of the PDEs compared with the state-of-the-art neural operators, which is very impressive. The proposed method also shows the ability to find the solution of a high-resolution input after learning from lower-resolution data.

**Limitations And Societal Impact:**

Authors should address the limitation of the filter matrices used in this paper. Should they apply to all PDE problems? Why does this approach result in 10x accuracy improvement for some PDEs while only gives about 1x accuracy improvement (no improvement) for the Navier-Stokes (NS) are 2d time-varying PDEs?

**Main Review:**

This paper will have a significant impact in the field of operator learning for differential equations. I am looking forward to its publication with some modifications to address the below issues:

1.	On Line 115 Page 3, should it be V(0, k) instead of V(0, n)?

2.	In Figure 1 on Page 4, the description for this figure is not clear. It is hard to understand (ii) and (iii) by the caption. It doesn’t even mention (iii) explicitly. What is the relation between H(0), G(0), H(1), G(1) the 3d matrix displayed in the figure? On the last line of the caption, it should be i = 0, 1 instead of i = 1, 2.

3.	On Line 178 Page 5, it is not clear what the predefined filter matrices look like. Only looking at Figure 2 and the contexts in the 2.3 Multiwavelet-based Model, it is very hard for the readers to build the same model on their side, hence it is hard to reproduce the experimental results in the paper. The authors should plot another graph for the neural networks so readers can understand how to build the neural network in practice. The author should show at least one detailed neural network for one of its PDE experiments for readers to reproduce the results.

4.	I don’t see the supplementary materials.

In summary, I recommend its publication in NeurIPS after those commends are addressed.


**Time Spent Reviewing:**

6

---

> ### Author Response · Authors · 2021-08-10
> **Response to reviews**
>
> We thank the reviewer for appreciating our results and the kind words! More importantly, we appreciate a careful read of the paper. Specifically, we will make the following minor revisions in the manuscript to take care of the pointed-out typos:
>
> 1. The typo would be corrected to $V_{n}^{k}$.
>
> 2. The caption of Figure-1 would be made more clear. We will correct the typo in ‘i=1,2’ and also insert the reference to (iii) subfigure. To clarify, the sparse 3d matrices are essentially the constituent terms for multiwavelet projection of the kernel from eqn (10). For each sub-matrix indicated by $A_n, B_n, C_n$ in Figure-1, they are kernel projections as written in lines 153-154 of the paper. According to eq. (6)-(9), the scale/wavelet coefficients can also be obtained from the multiwavelet filter matrices $H^{(i)}, G^{(i)}$ at higher/lower scales.
>
> 3. We have provided the complete set of computer codes for reproducing all of the experiments for future readers. The code is provided in the supplementary materials. The detailed algorithm to obtain the filter matrices is also provided in the uploaded code.
>
> 4. The supplementary materials are uploaded along with the main paper. We refer the reviewer to “WaveletPDE_Supplementary.pdf” for the full version of the paper, in which the Appendix begins from Page-15 onwards.
>
> **Limitations**: We have discussed the numerical issues (Checklist 1(b)) that we encountered with the current implementation of the filter matrices in the Supplementary materials (Section C.4) Page-25. The non-linear equations like Navier-Stokes are difficult to approximate using canonical integral kernels (with multiple layers and non-linearity) and require more data for better performance. Multiwavelets can be very well extended to derivative and exponential (Ref [4] of the paper) operators, and therefore, a principled future direction is to develop multiwavelets representation of non-linear operators which could better model such equations. By doing so, it would be possible to design an operator map that could potentially work with fewer data and giving better performance. We are excited to pursue this direction in follow-up work!

---

> > ### Comment · Reviewer_aK5Q · 2021-08-21
> > **Thanks for the clarifications**
> >
> > I keep my rating

---

### Official Review · Reviewer_z5a1 · 2021-07-11

**Rating:** 7
**Confidence:** 4

**Summary:**

This paper proposes to use a multiwavelet representation in order to learn operators which compute non-linear PDEs. The input and output functions are represented in multiwavelet bases.  The key idea is that multiwavelets provide sparse representation of large class of Calderon-Zygmund operators with a "non-standard form" which keep both the wavelet and scaling function coefficients, and are thus used in numerical schemes to solve many PDEs such as Burger equations or KdV.  In these cases, the specific non-linear operator corresponding to a PDE is a non-linear map between two sparse representations and are thus expected to be easier to learn then from the original time or spatial functions. The operator learning is performed with neural networks applied at each scale. The algorithm is resolution independent because the resolution parameter is incorporated in the multiwavelet basis. Results are demonstrated over Burger, Korteweg-de Vries and Darcy's flow with better than state of the art results.

**Limitations And Societal Impact:**

Yes

**Main Review:**

This paper is presenting an interesting approach to learn non-linear operators arising from PDEs. By using prior information on the nature of the solutions and the underlying operators, it  builds a sparse representations which simplifies the estimation problem. This type of work at the interface between harmonic analysis representations of PDEs solutions and neural networks is interesting and can potentially lead to real applications. At this point, the problems that is studied are very academic (one-dimensional Burger or KdV or Darcy flows) but they are none the less interesting. Whereas the paper spends a lot of time on the multiwavelet representation and its properties to represent Calderon-Zygmund integral operators, which are known in harmonic analysis, they spend little space on the description of the neural networks that perform the learning, the choice of network and an analysis of its properties. It would be very useful to see how the complexity of the network relates to the complexity of the underlying learned operator once represented in a sparse non-standard form. Overall this is a good paper that makes an interesting link between PDE and learning.

**Time Spent Reviewing:**

4h

---

> ### Author Response · Authors · 2021-08-10
> **Response to reviews**
>
> We are thankful that the reviewer has found our paper’s approach interesting! We also agree with the reviewer that the evaluation datasets are more on the academic side. But by exploring a wide spectrum of possible examples, multi-dimensional, multi-disciplinary ranging from fluid dynamics (Darcy flow), viscous fluids (Navier-Stokes), gas dynamics and traffic flow (Burgers equation), beam load (Euler-Bernoulli), waves on the water surface (KdV), we show that the proposed methods of multiwavelet operator learning seem a promising approach. We are excited to pursue the real-world datasets in follow-up work! Future work will look into its application to datasets, for example, aircraft wing sensor data, neuron-neuron interactions time-series data.
>
> The current work mostly focuses on the implications/advantages of the multiwavelet filters in the sparse learning of the integral kernel, and i.e. why we have used very simple single-layered CNNs for the A, B, and C NNs. The proposed model complexity is *input size-independent*, because of the non-standard form of multiwavelets, thus enabling us to ‘reuse’ each canonical NNs (A, B, C) at every scale. Therefore, the overall complexity of the MWT based operator learning model scales linearly with the complexity of the canonical NNs and not with the input resolution.
>
> It is very likely that complex NNs with multi-layered structures will result in more accurate predictions using our proposed MWT model, and that can be easily done with the current formulation. We left this as a future work, where for more complicated datasets a better choice of CNNs or clever DNN architecture is useful for A, B, and C NNs in Figure-2.

---

> > ### Comment · Reviewer_z5a1 · 2021-08-26
> > **Answer to authors**
> >
> > Thank you to the authors for their answer, I keep my rating.

---

### Official Review · Reviewer_RXFA · 2021-07-13

**Rating:** 6
**Confidence:** 4

**Summary:**

This paper studies the problem of approximating the solution operator of a partial differential equation (PDE) that maps the inputs to the solution. The authors begin by reviewing the deep learning techniques that have been recently introduced and the multiple applications of this problem. The main contribution of the paper is to propose a new neural network architecture, based on the multiwavelet transform, to approximate the kernel of the solution operator. Finally, the authors compare their method against several Neural operator techniques (Fourier neural operator, graph neural operator, ...) on a number of numerical PDE examples and achieve lower relative error (2X - 4X).

**Limitations And Societal Impact:**

I do not see any negative societal impact in this study. One limitation that the authors could discuss is that the method currently works on simple tensor-products domain, on which simple polynomial bases exist. I do not know if the method naturally extends to more complex geometries or non-uniform spatial discretization.

**Main Review:**

## Originality

The problem of approximating the solution operator is not novel and many techniques have been recently proposed, which the authors reference properly in the introduction. A deep learning approach based on the wavelet transform has also been proposed by [1] and I believe that the authors essentially generalize this approach to employ an orthonormal polynomial basis (Legendre or Chebyshev basis) as basis for the wavelet construction. The paper would gain in quality by explaining the advantages and drawbacks of this approach compared to [1]. The dimension of the subspace generated is larger, which gives higher approximation power, but does it come with higher computational training time for the networks ?

[1] Feliu-Faba, Fan, Ying, J. Comput. Phys., 2020.

## Clarity

The introduction of the paper is well-written and summarizes the potential applications as well as the recent works on the topic. However, I found several places where the paper lacks clarity, beginning with section 2. In the problem setup, the problem is not clearly defined and I do not see what the function a(x) represents for the PDE: is it a forcing term, a coefficient function in the PDE ?
Similarly, the authors "do not put the restriction of linearity" on the operator T, but represent it as an integral kernel and try to approximate the kernel. This formulation won't hold for the nonlinear PDEs that the authors are considering later.

Section 3.4 discusses standard properties of integral kernels but the definition and property cited are not mathematically rigorous as they omit the definition of the objects or the spatial domain. Furthermore, Property 1 does not hold for the Green's function of the Laplace operator in spatial dimension 3 (the kernel is singular and not continuous along the diagonal as stated by the property).
The experimental setup at the beginning of section 3 also lacks clarity: do the authors learn the operator for 1 input-output pair (a(x),u(x)), evaluated at N points of the domain (i.e. one-shot learning), or do they choose several input-output pairs, which they then sample on the domain ?

Most of the section A of the supplementary material is very standard and the authors could just reference the relevant results later instead of redefining Legendre, Chebyshev polynomials or describing Gram-Schmidt orthogonalization.

## Quality

The authors propose to use a compressed representation of the integral kernel (or Green's function) because learning a direct representation of the kernel is expensive in high dimensions (essentially because the kernel has twice the spatial dimension of the solutions to the PDE).

 However, it is feasible to learn the kernel for the problems that the authors are considering in the paper (spatial dimension 1,2, and simple square geometries) [2,3]. Moreover, even for simple problems (Poisson equation in 1D/2D), the kernel can be challenging to learn due to the singularity along the diagonal. I'm wondering whether the compressed approach discussed in the paper would be able to accurately learn the Green's function in this setting and outperform the methods proposed by [2,3] by comparing the L2 norm of the kernel function.
In  fact, using a testing set of functions to benchmark the method can be misleading, especially if the testing set is generated from the same distribution as the training set, because the method could simply overfit the training functions and not be accurate for any type of input functions.

Similarly, the choice of the Gaussian random fields for the input functions (e.g. $f \sim \mathcal{N}(0, 7^4(-\Delta+7^2 I)^{-2.5}$)) seems a bit strange and varies across the different examples. Since the method is also assessed on functions generated by the same distribution, it is also a possibility that the method performs well for this particular choice but fails for any other choice.
Why did the authors not consider the standard squared-exponential covariance kernel ?

Finally, the benchmarks are reported on an average of 3 experiments, but the authors did not report the individual errors, which questions the stability of the technique with respect to multiple runs and the standard deviation of the errors achieved.

[2] Li et al., arxiv:2003.03485, 2020.

[3] Boulle, Earls, Townsend, arxiv:2105.00266, 2021.

## Significance

The gain in the relative error (2X-4X) achieved by the author is moderately significant but is compensated by the different flaws in the numerical experiments pointed out in the previous section. This neural network architecture could be of practical interest, provided it is either computationally faster than existing techniques or more accurate (but validated by more robust numerical experiments). The theoretical justification of the method is a bit vague due to the lack of mathematical rigour in sections 2.1-2.4 and probably does not apply for the range of examples that the authors are considering.

**Time Spent Reviewing:**

4 hours

---

> ### Author Response · Authors · 2021-08-10
> **Authors clarification and response to reviews**
>
> We thank the reviewer for evaluating our work and providing valuable feedback. We now take the opportunity to clarify the concerns raised by the reviewer:
>
> The work in Feliu et al. (Ref [26] in our manuscript) has proposed an operator learning scheme for PDE of the kind $\mathcal{L}_{a}u = f$, where the map is learned from $a->u$. Our work is different in several angles from this work, specifically,
>
> (i) [26] needs the knowledge of the forcing function ‘f’ in the dataset along with the input ‘a’ and the output ‘u’, while our work learns a PDE agnostic map with only input ‘a’ and output ‘u’. The case of learning a map from PDE coefficients ‘a’ to solution ‘u’ is a special case that we have considered in Sec-3.4.
>
> (ii) [26] is not a resolution-independent scheme and the size of the model grows with the input resolution, while our approach is *resolution-independent* and the model size is input resolution invariant.
>
> (iii) [26] needs to learn the wavelet filter from the data, while in our work the multiwavelet filter matrices (H, G) are *fixed* beforehand, thus saving the training efforts. We show the importance of carefully computed filter matrices H, G in Table-2 where RND matrices perform worst.
>
> (iv) Our work by generalizing the multiwavelets to arbitrary measures allows developing a series of models, while [26] has only one model using the wavelet decomposition.
>
> (v) Although our work has filter matrices that are fixed thus saving the training efforts, due to the unavailability of source code of [26] it is difficult to compare the actual run-time.
>
> We have introduced the operator map $Ta = u$ as a general problem in Sec-2 and then discussed its specific applications in the context of PDEs. As demonstrated in the Results section-3:
>
> (i) In Sec-3.1, 3.3, we set up the operator map problem as T u(x,0) = u(x,1), where u(x,0) is the initial condition and u(x,1) is the solution at time t=1 (line 243-244, and line 290-291).
>
> (ii) In Sec-3.2, we set up the operator map problem as T f(x) = u(x), where f(x) is the forcing function and u(x) is the fourier transform of time-varying beam displacement (line-275-280).
>
> (iii) In Sec-3.4, we set up the operator map problem as T a(x) = u(x), where a(x) are the PDE coefficients and u(x) is the solution (line 316).
>
> For non-linear operator maps, the linearity assumption over the kernel does not hold as the reviewer has also pointed out, for example, a map from an initial condition to the final solution can be formulated as an exponential operator. As mentioned in Sec-3 ‘Model architectures’ we have used a multi-layered MWT cell with ReLU non-linearity to be able to learn the non-linear maps with each layer being a canonical MWT cell (Figure-2). Further, as mentioned in line 211-213, each data pair is $(a_i, u_i)$, where $a_i/u_i$ are function evaluations at M-discretized locations of D. There are a total of N input/output pairs $(a_i, u_i)$ in the dataset evaluated using $x_1, x_2, ..., x_N$ each of which is a M-point evaluation. Therefore, we choose several input/output pairs for training, each sampled at M locations which is the same setting as used in the previous neural operator works [42-44].
>
> **Property-1** The Property-1 from [17] is a simple outcome for pseudo-differential operators over univariate functions, i.e. $D = \mathbb{R}$. The reviewer is correct that the green function of the 3d Laplace operator is not $C^{0}$ at the diagonal but for the 1d Laplace operator, it is $C^{0}$. We will add a sentence in line-195 to clarify that Property-1 is for the 1d domain. Note that the Euler-Bernoulli equation in Section 3.2 is a one-dimensional PDE.
>
> **Kernel estimation** From the perspective of the operator map learning, learning the green function is also important but then is limited to linear PDEs. On the other hand, the proposed MWT models can be PDE agnostic, and learn the map not only from forcing function to PDE output (Sec-3.2) but also from initial conditions to a solution at a later time (Sec-3.1, 3.3) and PDE coefficients to PDE solution (Sec-3.3). The reviewer has also agreed that estimation of the kernel can be tricky due to the presence of singularities as we have also mentioned in the paper, but to the advantage of multiwavelets, as long as the singularities are finite (Sec-2.4), then the kernel has an efficient representation in the multiwavelets domain. Further, due to the *vanishing moments property* (Eqn (5)) of multiwavelets, such kernels (for example, Calderon Zygmund) have compressed representation through multiwavelets. In the future, the kernel estimation problem can also be explored using multiwavelets. While we have already cited Li et al. 2020, we would be happy to cite Boulle et al. 2021 such that our work reaches a broader audience.
>
> **KdV equation** For the KdV equation, we have used random functions from Gaussian Random fields (GRF) $f\sim\mathcal{N}(0, 7^4(\Delta+7^2I)^{-2.5})$ with periodicity for the results in Table-1. Such GRF is useful because of the nature of the KdV equation modeling shallow-water waves. The GRF with finite Fourier series coefficients produces smooth random functions with periodicity to mimic the required physical behavior. Apart from this GRF, we have already used a different function generating scheme [27] in Figure-4 where we control the fluctuation strength $\lambda$ of the input function to show the improvements offered by the MWT model compared to the recent successful FNO [42]. In addition, as per the reviewer’s suggestion, we also sample the random function from the squared exponential kernel (SEK) [Chap 4, R3.1] for the KdV equation. We have used the periodic version of the SEK
>
> $k(x, x’) = \exp{(-2\frac{\sin^{2}(\pi(x-x')/P)}{L^2})}$,
>
> where, P = 1 (domain length), and L = 0.5 is chosen to avoid sharp peaks in the random functions. For the same setting as in Table-1, for s = 64, 128, 256, 512, 1024 we get an average relative error of 0.00228, 0.00247, 0.00200, 0.00238, 0.00224 for MWT Leg. This is comparable to the existing presented result with GRF in Table-1. The use of a single generating rule to obtain the complete dataset and then obtaining train/test functions is used in all of the recent successful neural operators (FNO [42], GNO [43], MGNO [44]) that we have compared. To make a fair comparison, we have used the existing datasets with the same train/test samples. However, we have also followed the reviewers’ suggestion and run additional experiments where we sampled train/test from different generating rules. We go a step further and show the importance of fluctuation strength in addition to the results of Figure-4 in this experiment. Specifically, we have observed that compared to having different generating schemes for train/test, the difference in fluctuation strength of train and test data matters more. It is intuitive that training only with low fluctuating functions will have adverse effects when trying to predict high fluctuating samples. In our experiments, we train on periodic SEK as discussed above, but test on data obtained from a different generating scheme (Ref [27] of our paper) with a controllable parameter $\lambda$ (similarly in Figure-4).
>
> |  | $\lambda = 0.25$ | $\lambda=0.5$ | $\lambda=0.75$ |
> |--- | --- | --- | --- |
> |s = 256 | 0.009484 | 0.006950 | 0.002905|
> |s = 1024 | 0.010303 | 0.006210 | 0.002539|
>
> We see that when the test data has higher fluctuations compared to the train data, the performance degrades. This is a useful result, in addition to what we have in Figure-4. In particular, it is helpful for designing better application-specific neural operators, as well as, to carefully select training data to make better predictions in the future. Overall, we show that the proposed operator learning is working for the multiple set of generating rules, as well as different rules across train/test.
>
> For Table 1, the standard deviation of the results for MWT Leg are 0.0001138 (s=64); 0.0000756 (s=128); 0.0000548 (s=256); 0.0000705 (s=512);  0.0000567 (s=1024). For Table 2, the standard deviation of the results for MWT Leg are 0.0001288 (s=32); 0.0000361 (s=64); 0.0000734 (s=128); 0.0000953 (s=256). We have observed that after 400 epochs the performance is very stable, and we trained for 500 epochs as mentioned in the paper; the error results with standard deviation can also be provided in the full version of the paper with supplementary materials.
>
> The multiwavelets analysis of operators can be extended to other forms, for example, derivative and exponential operators (in addition to the considered integral kernels). This is an interesting research direction for operator map learning problems to tackle complex scenarios. Non-uniform discretization is also a useful addition to all of the neural operator works which should be addressed in the future.
>
> [R3.1] C. E. Rasmussen, C. Williams, Gaussian Processes for Machine Learning (MIT Press,
> 2006).

---

> > ### Comment · Reviewer_RXFA · 2021-08-23
> > **Response to the authors**
> >
> > Thanks to the authors for the detailed response to my comments.
> > The clarification with respect to the prior work and the additional experiments conducted seem convincing to me and I have updated my rating to recommend acceptance.

---

> > > ### Author Response · Authors · 2021-08-25
> > > **Thank you!**
> > >
> > > We are glad to see that the reviewer's concerns are addressed and would like to thank for the updated score. We are working towards the suggested experiments and a complete set of additional results will be updated in the revised paper.

---

### Official Review · Reviewer_SQX4 · 2021-07-15

**Rating:** 7
**Confidence:** 3

**Summary:**

This paper presents a multiwavelet-based neural operator learning scheme that compresses the associated operator’s kernel using fine-grained wavelets. Using the proposed model, the paper performs experiments on the Korteweg-de Vries (KdV) equation, Burger’s equation, Darcy Flow, and Navier-Stokes equation. Compared with the existing neural operator approaches, this model claims to show significantly higher accuracy and can find the solution of a high-resolution input after learning from lower-resolution data. This paper appears to be a novel proposal, and certainly of interest. However, the method only applies to some toy examples, its application prospect seems narrow.


**Limitations And Societal Impact:**

I would love to see some limitations of the proposed method. In addition, the method should discuss more robustness to real noise data.

**Main Review:**

Strengths


This paper operates in a general area (neural PDE) of great interest to the machine learning community. The method demonstrates the applicability on higher dimensions of the 2-D Darcy flow equation in Section 3.4, which is a good point. Also, I appreciate the detailed supplementary material provided by the authors.

Weaknesses


- The introduction of the theory seems a little indirect and not concise, such as Section 2.1 is a bit redundant.
- Although the paper shows a lot of error curves, the results are rarely shown. To enhance readers' intuition of the predictive power of the presented model, the temporal evolution of the PDE solutions should be visualized.
- The result comparison is not convincing in its current form. It's described in a fairly vague way, and there's a lack of results showing that it works. This part would need significant extensions, or potentially could be left out.
- Table 3 in supplementary material looks like the simulation error is too large to be practical.


Correctness


The technical content of this paper appears to be correct.


Clarity


Overall, the exposition of the paper is fairly clear. But some details are confusing:
- MWT model architecture in Figure 2 is confusing. I can't follow the data flow of the network.
- What are the two subfigures in Figure 3?  Please specify the meaning of these two figures.



Relation to prior art


This paper compares its MWT model using two different OP bases (Leg, Chb) with the most recent successful neural operators, such as the graph neural operator (GNO), the multipole graph neural operator (MGNO), the LNO which makes a low-rank representation of the operator kernel K(x; y), and the Fourier neural operator (FNO ). The MWT model claims to show the ability to learn the function mapping through lower-resolution data, and be able to generalize to higher resolutions. These methods are compared in the results section, but the demonstration seems to need to be strengthened. Also, it would be helpful to add some citations of recent Neural ODE/PDE solvers, such as papers related to conservation laws, symplectic neural networks, and neural vortex methods.

Reproducibility


The method description is clear in general and reasonably easy to follow.



Additional Feedback/Questions


How to handle the modal decomposition of a problem with nontrivial boundary conditions?





**Time Spent Reviewing:**

6

---

> ### Author Response · Authors · 2021-08-10
> **Response to the reviews**
>
> We thank the reviewer for acknowledging the novelty of our results. While we have compared against the benchmark PDE equations that all of the recent successful neural operator works have used, in addition, we have also explored PDEs like the KdV and Euler-Bernoulli PDEs to set up a better understanding of the operator map learning behavior. By exploring a wide spectrum of possible examples, multi-dimensional, multi-disciplinary ranging from fluid dynamics (Darcy flow), viscous fluids (Navier-Stokes), gas dynamics, and traffic flow (Burgers equation), beam load (Euler-Bernoulli), waves on the water surface (KdV), we show that the proposed methods of multiwavelet operator learning seem a promising approach. We are excited to pursue further applications as well as real-world datasets in the future!
>
> The use of multiwavelets for dealing with operator learning by exploiting their fundamental properties can have huge implications for designing future models. While we have started the research with integral kernels, the work can be extended to *derivative and exponential operators* as well.
>
> **PDE solutions visualization**: The prediction of the current operator model is shown in Figure-3 (Right) for the KdV equation for an input Figure-3 (Left). In addition, we have also shown the prediction of the current model which is trained at a lower resolution to predict a higher resolution output in Supplementary materials Figure-10, 11 on Page-28. The temporal evolution of the predicted PDE solution for Navier-Stokes presented in the supplementary materials Section-D.1 can be visually produced by using the provided pre-trained models for $\nu=1e-3$ and $\nu=1e-4$ in ‘test_NS*.py’ uploaded along with the code package. For the reviewer’s and future reader reference, we are providing a visual representation [HERE](https://www.dropbox.com/sh/zd8lhjqea9yp0qk/AAAXj3ZZtGlv1Nxp4tJ4e1Ffa?dl=0).
>
> In Table-3, we see that for a lower viscosity coefficient $1e-3$, even 1000 training samples are sufficient for getting a very good performance ~0.006 from a non-linear time-varying Navier-Stokes equation. But as we decrease the $\nu$ which then implies the possibility of turbulence, the 1000 samples are not sufficient and more samples $N=10000$ are required to get a lower error ~0.066. Similar behavior is observed when we further reduce the $\nu$ to $1e-5$ where 1000 samples are not sufficient. We note that non-linear NS equations can be modeled as exponential and derivative operators. By doing so, a multiwavelet representation of such an operator could be explicitly derived (Ref [4] of the paper). This is an interesting research direction for the future, where potentially better performance could be obtained even with fewer training samples.
>
> **MWT cell description**: The MWT model in Figure-2 is presented in the form of a recurrent cell. The input data at each iteration is $s^{(n+1)}$ which then gets transformed into $s^{(n)}, d^{(n)}$ using the filters $H^{(i)}, G^{(i)}$. At the same iteration, we also obtain the corresponding outputs $U_d^{(n)}$ and $U_{\hat{s}}^{(n)}$. The same process is repeated in the next iteration but now with using $s^{(n)}$ (obtained from the previous step) as the input, thus this makes a recurrent chain of operations and is a kind of ladder-down operation. The loop is repeated till we reach the L-th scale at which the final operation of $\bar{T}$ is applied which is according to equation (10). Finally, in the reconstruction step (which is ladder-up) the outputs $U_d^{(n)}$ and $U_{\hat{s}}^{(n)}$ are combined using a reconstruction filter to iteratively obtain the finer scales, and finally, $U_s^{(N)}$, N is the finest scale of the output. Additional space permitted, we would like to add a detailed description with examples in the main paper as well for Figure-2.
>
> The subfigures in Figure-3 are the input $u_{0}(x)$ on the left, and the output $u(x,1)$ on the right as explained in Section 3.1. We would also like to modify a little the current caption of Figure-3 to better explain the figures as follows:
>
> Figure-3: **The output of the KdV equation**. (Left) An input $u_{0}(x)$ with $\lambda = 0.02$. (Right) The predicted output of the MWT Leg model learning the high fluctuations.
>
> **Additional references**: We thank the reviewer for suggesting to include references related to neural ODEs, symplectic neural networks, and neural vortex methods. We will be adding the following references to our paper such that our work can reach a broader audience.
>
> [R2.1] Kidger et al., Neural Controlled Differential Equations for Irregular Time Series, Neurips 2020
>
> [R2.2] Z. Chen et al., Symplectic Recurrent Neural Networks, ICLR 2020
>
> [R2.3] S. Xiong et al., Neural Vortex Method: from Finite Lagrangian Particles to Infinite Dimensional Eulerian Dynamics, arXiv:2006.04178
>
> **Additional comments** While the modal decomposition techniques like proper orthogonal decomposition (POD) and dynamic mode decomposition (DOD) have applications in getting a low-dimensional approximation of PDEs, it is out of scope to discuss its implications in the current work which is more towards learning an operator map. We would also like to mention that our work by using multiwavelets has no restrictions on periodic boundary conditions, and non-periodic versions like Darcy flow and time-varying Navier Stokes are also considered.
>
> **Limitations**: We have discussed the numerical issues (Checklist 1(b)) that we encountered with the current implementation of the filter matrices in the Supplementary materials (Section C.4) Page-25.

---

> > ### Comment · Reviewer_SQX4 · 2021-08-16
> > **Update**
> >
> > Thanks for your response. Most of my concerns are addressed. I hope the authors can improve the method description in the next submission to make the paper clearer and easier to follow.

---

> > > ### Author Response · Authors · 2021-08-17
> > > **Thank you!**
> > >
> > > We are happy to have addressed the reviewers' concerns. The clarity of the methods will be improved in the revision.

---

### Official Review · Reviewer_rY49 · 2021-07-16

**Rating:** 8
**Confidence:** 4

**Summary:**

The work develops Multiwavelet-based neural operator using multiwavelet Transform and orthogonal polynomials. The model applies multiscale transfer and kernel integration. Empirically the model is more accurate compare to Fourier neural operator on many PDEs.

**Limitations And Societal Impact:**

It will be better if the author can discuss when wavelet does not apply. To my knowledge, while there are many works on wavelets, but in general people still prefer FEM/FDM solvers, etc.

**Main Review:**

In my opinion, this work is a natural and novel extension to neural operators. The architecture is very solid and the performance is quite good. I would vote for strong acceptance.

Concerns:
(1) It's understandable that most of the benchmark numbers are copied from the previous papers. For the Burgers' equation, it was reported as 0.0149 on FNO but if the batch norm layers are taken out, the error on FNO will be ~ 0.002. This can also be true for the Korteweg-de Vries equation. This may explain why the gap of MWT and FNO is so larger on Burgers and Korteweg-de Vries, but not so much on Darcy and Navier-stokes.
To have a fair and solid comparison, I strongly suggest the author use the tuned FNO model and re-run for Burgers and Korteweg-de Vries equation.
(2) MWT has a multi-scale structure, which may imply the overall runtime of MWT can be slower. It will be great if the author could report the runtime of MWT and how it compares to the baseline models.

**Time Spent Reviewing:**

3

---

> ### Author Response · Authors · 2021-08-10
> **Response to the reviews**
>
> We thank the reviewer for evaluating our work, and it is encouraging to see that the reviewer has found our work “strong”!
>
> **Tuning FNO** As pointed out, we have taken the Burgers, Darcy flow, and Navier-Stokes performance data for existing neural operators from the recent successful FNO work [42]. While for the KdV equation, we ran the FNO with the open-source code for 1D data, we can confirm that it does not have any Batch normalization layers. Since we have borrowed the results of Burger's equation from the original FNO paper, we agree with the reviewer that an additional result FNO (fine-tuned) be added to the existing results in the revision. We also wish to point out that on a similar note, fine-tuning of the MWT based model could also be done in which the currently used simple single-layered CNNs for A, B, and C could be changed to have better results.
>
> In the author's opinion, the primary motive is to put forward the multiwavelet-based analysis of the operator kernel for efficient learning, as well as to motivate the community for future research to explore derivative, exponential operators (in addition to integral kernels) for the complex real-world situations.
>
> **Efficient implementation** The MWT has multi-scale functionality but at each step, the dimension of the data is reduced to half equations (6)-(7). Theoretically speaking, this results in $\mathcal{O}(Mlog(M))$ complexity with M as the input size. One issue, however, is the efficient implementation. The Discrete Fourier transform enjoys an efficient parallel implementation using the FFT algorithm, while no such architecture (to the best of the authors’ knowledge) is explored for the multiwavelet transform. As an example, the average training time/epoch (over 3 experiments) for the KdV equation for s = 64 is FNO:0.8457sec, MWT Leg:1.8103sec and for s=256 we have FNO:0.8927sec, MWT Leg:2.3492sec.
>
> We wish to point out that the multiwavelet filters H, G have a *sparse structure*, see Page-23 for Legendre and Page-25 for Chebyshev in the supplementary materials. In particular, $H^{(0)}, H^{(1)}$ are lower-triangular matrices that are reminiscent of the Gaussian quadrature for Legendre polynomials. For OPs with uniform measure (Legendre), we will also have sparsity in the $G^{(0)}, G^{(1)}$ as they would be upper-triangular matrices. Using this information beforehand, the filtering operations in equations (6)-(9) can be efficiently implemented to avoid a full matrix multiplication. Although not considered in the current work, efficient implementation of multiwavelets in signal processing using matrix spectral decomposition is explored in [R1.1], recursive block matrices to construct semi-orthogonal multiwavelets in [R1.2]. It is left for future work to explore the possibility of efficient implementation of multiwavelets in deep learning architectures.
>
> **Limitations**: We have discussed the numerical issues (Checklist 1(b)) that we encountered with the current implementation of the filter matrices in the Supplementary materials (Section C.4). Wavelets provide an efficient operational matrix formulation to solve PDEs, but they also require a large number of coefficients to make the functional approximation (Ref [16, 33, 36, 57] of the paper). Classically, the discrete wavelet transform is built to deal with functions defined over the entire real line, and for finite interval functions, there are well-known 'boundary issues’ which puts a limitation on its general use. There is rich literature dealing with boundary issues and designing application-specific boundary functions. Since we dealt with only the transformation of scale/wavelet coefficients in our model eqn (6)-(11), we have not encountered this issue in our work.
>
> [R1.1] Kolev, V. et al. “Design of a Simple Orthogonal Multiwavelet Filter by Matrix Spectral Factorization.” Circuits, Systems, and Signal Processing 39 (2020): 2006-2041.
>
> [R1.2] M. Cotronei et al., "Multiwavelet analysis and signal processing," in IEEE Transactions on Circuits and Systems II: Analog and Digital Signal Processing, vol. 45, no. 8, pp. 970-987, Aug. 1998.

---

> > ### Comment · Reviewer_rY49 · 2021-08-24
> > **concerns addressed**
> >
> > Thank the authors for the response. My concerns are addressed. Looking forward to see the updated results of the FNO baseline if it is available.

---

> > > ### Author Response · Authors · 2021-08-25
> > > **thank you!**
> > >
> > > We are thankful to the reviewer for raising the computational points and are happy to see that the concerns are addressed. Indeed, a better computational approach for multiwavelets is a promising research direction. Finally, we are in the process of running the suggested experiments by all of the reviewers and the complete set of results will be updated in the revised version soon.

---

### Author Response · Authors · 2021-08-10
**Authors response**

We thank all the reviewers for their careful reading and thoughtful comments/suggestions on our paper. We find it encouraging that reviewers have found our work interesting and novel! The reviewers' suggestions have led us to further improve the clarity of the manuscript as well as to add more technical details. We have addressed the individual reviewer’s comments below and here we summarize the proposed changes.

1. **MWT architecture description**: Adding a detailed description of the Figure-2 MWT model to better explain the architecture.
2. **Additional experiments**: We will add the fine-tuned FNO for Burgers equation in the existing results. As the supplementary results, we will also add the KdV equation with the squared exponential kernel generating scheme as well as train/test with different generating rules.
3. **Additional References**: A variety of additional references will be added to the revised version such that our work reaches a broader audience.
4. **Figure Captions**: The captions of Figure-1,3 will be updated to further clarify the existing visuals.

---

### Decision · Program_Chairs · 2021-09-27

**Decision:**

Accept (Spotlight)

**Comment:**

This paper investigates the problem of approximating the solution operator of a partial differential equation (PDE) and proposes a novel neural architecture based on the multi-wavelet transform to approximate the kernel of the solution operator. The key innovation of this paper is a clever application of the multi-wavelet transform that allows for a simpler solution. There is a general consensus among the expert reviewers that this paper is of high quality and offers a new solution to the important problem that would have a high impact on the PDE community. Some reviewers raise a criticism that the paper didn't provide the results on complex real-world scenarios but focus instead on the simple settings. I consider this weakness to be minor for the NeurIPS audience, especially after the authors have satisfactorily described potential directions to improve this work in their responses.

As a result, I recommend this paper for publication at NeurIPS2021.